# Extracellular Vesicles and Viruses: Two Intertwined Entities

**DOI:** 10.3390/ijms24021036

**Published:** 2023-01-05

**Authors:** Coline Moulin, Mathieu J. F. Crupi, Carolina S. Ilkow, John C. Bell, Stephen Boulton

**Affiliations:** 1Ottawa Hospital Research Institute, Ottawa, ON K1H 8L6, Canada; 2Biology Master at École Normale Supérieure of Lyon, Claude Bernard Lyon 1 University, University of Lyon, CEDEX 07, 69342 Lyon, France; 3Department of Biochemistry, Microbiology and Immunology, University of Ottawa, Ottawa, ON K1H 8M5, Canada; 4Department of Medicine, University of Ottawa, Ottawa, ON K1H 8M5, Canada

**Keywords:** extracellular vesicles (EVs), virus-host interactions, virotherapy, cancer, gene therapy

## Abstract

Viruses share many attributes in common with extracellular vesicles (EVs). The cellular machinery that is used for EV production, packaging of substrates and secretion is also commonly manipulated by viruses for replication, assembly and egress. Viruses can increase EV production or manipulate EVs to spread their own genetic material or proteins, while EVs can play a key role in regulating viral infections by transporting immunomodulatory molecules and viral antigens to initiate antiviral immune responses. Ultimately, the interactions between EVs and viruses are highly interconnected, which has led to interesting discoveries in their associated roles in the progression of different diseases, as well as the new promise of combinational therapeutics. In this review, we summarize the relationships between viruses and EVs and discuss major developments from the past five years in the engineering of virus-EV therapies.

## 1. Introduction

Multicellular organisms rely on intercellular communication to regulate many aspects of their physiology. It defines environmental niches that regulate cell growth and behavior, and it is essential for collective defense against host pathogens. The majority of intercellular communication is mediated via the transportation of bioactive molecules, such as proteins, nucleic acids, metabolites and lipids between cells [1,2]. Passage of these molecules can occur by passive diffusion or by transport via carrier molecules such as extracellular vesicles (EVs). EVs are cell-secreted membrane vesicles of various sizes, compositions and origins that induce physiological changes in recipient cells through the delivery of bioactive molecules. The biomolecules contained within EVs vary depending on the tissue of origin, immune set-point and cellular context [3] and web-based compendiums such as ExoCarta [4], Vesiclepedia [5] and EVpedia [6] are now used to document the vast array of biological molecules identified in EVs of different classes [1].

EVs are derived from multivesicular bodies (exosomes), the plasma membrane (microvesicles) [7] or other organelles such as the autophagosome [8]. Microvesicles range from 50 to 1000 nm in diameter and are released from the plasma membrane due to ATP-dependent contraction of the actin-myosin network [7]. Exosomes are EVs from 30 to 150 nm in diameter that are initially formed inside multivesicular bodies (MVB) and released after fusion of MVBs with the plasma membrane [7]. In contrast to exosomes, the secretion of microvesicles is dependent on cell signaling pathways [7]. Microvesicles and exosomes membrane composition share many characteristics, with the presence of tetraspanin (e.g., CD9, CD81, CD63), phosphatidylserine (PS) lipids, cell adhesion proteins (e.g., integrins) and intracellular trafficking proteins (e.g., Rab-GTPases, annexins) [7]. In addition, their uptake by recipient cells follows similar pathways and mechanisms [7]. Autophagosome-derived EVs are the result of a non-classical processing of the autophagosome. Instead of fusing with lysosomes, autophagosomes can fuse with endosomes or MVBs to form amphisomes or instead fuse with the plasma membrane. Both events eventually lead to the release of autophagosome-derived EVs [8].

EVs can stimulate immune responses against pathogens and tumors by transporting antigens and immune-stimulating factors, maintain cellular homeostasis by excreting harmful components, such as nuclear DNA in the cytoplasm [9,10] and play a role in pregnancy, stem cell differentiation and injury recovery [11]. For instance, EVs secreted by adipose mesenchymal stem cells are essential in the control of cell proliferation, migration, apoptosis, but also in angiogenesis, nerve regeneration and immune responses [12]. While many EV-regulated signaling pathways are beneficial to cellular homeostasis and host immunity, there are also many cases where EVs propagate or exasperate pathological conditions. EVs can transport misfolded amyloidogenic peptides associated with neurodegenerative diseases, such as Parkinson’s and Alzheimer’s disease [13], aid in the role of cancer progression by promoting cell proliferation, migration and angiogenesis [14] or transport apoptotic and inflammatory molecules leading to cell death and other inflammatory diseases [15].

EVs also have a deeply interwoven relationship with viruses [16,17]. Viruses can exploit EV pathways to benefit all aspects of their life cycle, from entry to egress and modulation of host immune responses [16,17]. Conversely, EVs can also be a powerful tool for alerting the body of viral infections and stimulating antiviral responses. The interplay between viruses and EVs is incredibly important for the regulation of viral pathogenesis. Nothing demonstrates this better than the duality of immune modulation through EV signaling in infected cells. EVs that normally alert surrounding cells to the presence of virus infection and stimulate antiviral responses can be hijacked by some viruses to instead downregulate immune responses in neighboring cells and make them more susceptible to infection [16,17].

In this review, we focus on the past decade of discoveries regarding the intertwined interactions of EVs and viruses. We begin by describing the biosynthetic pathway of EVs and discuss the ways in which viruses also utilize or manipulate these pathways for their own benefit. We examine the role of EVs in virus transportation and pathogenesis and discuss the relationships between EVs, viruses and the immune system. Lastly, we focus on the role of EVs in virus-associated disorders and discuss promising approaches for combining viruses and EVs as therapeutics, with a specific focus on cancer-related therapies.

## 2. Intertwined Intracellular Pathways of Viruses and EVs

### 2.1. EV Biosynthesis Pathway

EVs are secreted from intraluminal vesicles (ILVs) contained in cellular multivesicular bodies (MVBs). The biosynthesis pathway begins with the maturation of endosomes into MVBs when membrane cargos recruit factors inducing the internal budding of the endosomal membrane and the formation of ILVs. These cargos can include proteins, nucleic acids, metabolites and lipids [1,2]. Cargo uptake is governed by different mechanisms depending on the type of cargo and their interactions with EV packaging complexes.

Packaging of proteins into EVs depends on post-translational modifications—mostly ubiquitination and, in some cases, sumoylation [18]—as well as specific interactions with lipids and proteins inserted inside MVBs. The uptake of ubiquitinated proteins has been widely studied and is mediated by the endosomal sorting complexes required for transport (ESCRT) machinery. Ubiquitinated proteins interact with ESCRT-0, which recruits ESCRT-I, including the TSG101 subunit. ESCRT-I recruits, in turn, ESCRT-II and -III. Altogether, the ESCRT machinery mediates the invagination of MVBs and the scission of ILVs with the help of the vacuolar protein sorting-associated protein 4 (VPS4). Deubiquitylation is then required to sort proteins inside EVs (Figure 1A). In some cases, sumoylation also mediates protein uptake by the ESCRT machinery. For instance, Kunadt et al. showed that the sumoylation of α-synuclein is necessary for its interaction with the ESCRT complex and its packaging into EVs [19].

Non-classical sorting of proteins into EVs can also be achieved through ESCRT-independent pathways. ALIX can bind and transport non-ubiquitinated cargos inside MVBs, bypassing ESCRT-0, -I and -II [20]. ALIX is also recruited by the cell adaptor syntenin, which itself binds syndecan cargo inserted within the MVBs membrane. This syndecan-syntenin-ALIX complex mediates ILV formation in a partially ESCRT-independent pathway [33]. Moreover, proteins can be packaged inside ILVs through a sphingolipid-dependent, ESCRT-independent mechanism. MVB membranes contain sphingolipids, which can be hydrolyzed into ceramide by neutral sphingomyelinase-2 (n-SMase-2). Ceramides inserted inside the MVB membrane induce the negative curvature of the membrane and ILV formation [34]. Then, ceramide conversion into sphingosine-1-phosphate (S1P) enhances the activity of the inhibitory G protein-coupled S1P receptor, which mediates cargo sorting inside ILVs [35].

In contrast to protein cargos, the packaging of nucleic acids into EVs is less well understood but appears to depend on a variety of mechanisms. MicroRNAs (miRNAs) are packaged into EVs according to sequence motifs that are recognized by RNA-binding proteins, such as heterogeneous nuclear ribonucleoprotein A2B1 (hnRNPA2B1), synaptotagmin-binding cytoplasmic RNA-interaction protein (SYNCRIP), Argonaute2, Y-Box binding protein 1 (YBX-1), MEX3C, major vault protein (MVP) and La protein. These RNA-binding proteins then interact with MVBs membrane proteins, such as caveolin-1 (Cav-1), n-SMase-2 and VPS4, to facilitate the sorting of miRNAs into MVBs [36]. For instance, sumoylated hnRNPA2B1 binds specific sequence motifs (GGAG) in some miRNA and is then recruited by MVBs [37]. During cellular stress, Cav-1 is relocated from the plasma membrane caveolae to the MVB membrane leading to the hnRNPA2B1 uptake into MVBs [36]. The genomic DNA (gDNA) and mitochondrial DNA (miDNA) are also sorted into MVBs [38]. The damaged gDNA can be transported out of the nucleus inside unstable nuclear membrane vesicles called micronuclei, which eventually release the gDNA in the cytoplasm. Similarly, miDNA can leak in the cytoplasm during cellular stress. Then, free gDNA and miDNA are uptaken by MVBs [38] (Figure 1A).

Tetraspanin small transmembrane proteins also have a significant role in the EV biosynthesis pathway. Tetraspanins are known for their role in mediating many physiological processes, such as cell migration, adhesion and signaling. Various tetraspanins are located inside MVBs, such as CD63, CD9 and CD81 [11]. They support protein and nucleic acid cargo trafficking, selection and sorting into ILVs, and subsequent EV uptake by recipient cells [39]. For instance, CD63 found at micronuclei membrane can mediate gDNA sorting from the micronuclei into ILVs [38].

Once cargos are packaged into ILVs, MVBs either fuse with lysosomes to be degraded or move to the plasma membrane, where they fuse and release ILVs to the surrounding environment through processes regulated by the soluble NSF attachment protein receptor (SNARE) complex [21]. These secreted ILVs are termed EVs. The transport of MVBs to the plasma membrane is mediated by different Ras-related proteins in brain (Rab) GTPases [22]. Most notably, Rab-11 and -35 mediate ESCRT-independent MVBs secretion, while Rab-27 acts in an ESCRT-dependent MVB pathway [22]. However, the involvement of other Rabs in EVs biosynthesis pathway is still being discovered. For instance, it was found recently that Rab31 mediates cargo uptake into ILVs through flotillin proteins in an ESCRT-independent pathway and later inhibits MVB fusion with lysosomes to ensure EVs release in the extracellular environment [40] (Figure 1A).

Secreted EVs have been found in many body fluids, such as blood, urine, saliva, tears and semen [41], and their extracellular half-life depends on a variety of factors that are intrinsic to their composition, origin of production and current location [41]. However, some studies showed that EVs could only be detected in mice up to 30 min post-intravenous injection, suggesting a short half-life [41]. While EVs often spread to surrounding cells in the same tissue, it is also not uncommon for them to be distributed to distant tissues [11,42]. The selectivity of Evs for specific cell types can be determined by the cells they are produced from, and in vivo, this may even affect their accumulation in specific organs [11,42]. For instance, EVs secreted by mesenchymal stem cells are transported preferentially to the liver, the spleen and the kidney [11]. The factors within the stroma can also have a significant role in determining EV mobility and distribution. Lenzini et al. compared the ability of EVs derived from mesenchymal stem cells versus liposomes and polystyrene beads to escape confinement from alginate hydrogels [43]. They found that EV migration depends on the presence of aquaporin-1 inside their membrane, allowing water flow and EV deformation. In addition, they found that an extracellular matrix with an increased rigidity and stiffness enhanced EV mobility relative to a more elastic extracellular matrix, which could limit EVs access to some types of tissues [43].

Finally, EVs are internalized by specific recipient cells through EV-associated transmembrane molecules, such as extracellular matrix proteins (ICAM-1, laminin, fibronectin), integrins, proteoglycans, lectins, glycolipids, phosphatidylserine (PS) and tetraspanins [11]. EV binding to recipient cells can activate cellular pathways or lead to internalization through membrane fusion, clathrin-, caveolin- or lipid raft- endocytosis, but also through phagocytosis and micropinocytosis [3,11]. Interestingly, the uptake of EVs by recipient cells is enhanced by EV purity, smaller size, abundance and the local environment (e.g., pH, temperature) [3].

### 2.2. Viruses Hijack the EV Biosynthesis Pathway

Viruses have adapted strategies to hijack EV biosynthesis machinery to aid in all stages of their life cycles. Both RNA and DNA viruses can acquire their viral envelope by using the ESCRT complex and Rab-GTPases. Enveloped viruses can acquire their viral envelope while entering MVBs, as observed for the human cytomegalovirus (HCMV) [23], Human herpesvirus 6 (HHV-6) [44], SARS-CoV-2 [24], Dengue virus (DV) [45] and Hepatitis B virus (HBV) [25]. For instance, HBV large hepatitis B surface proteins (LHBs) hijack Rab5B, resulting in their transportation from the endoplasmic reticulum to the MVBs. There, LHBs recruit HBV capsid and TSG101 and bud inside MVBs [25]. Then, HBVs hijack Rab7a and Rab27, which enhance MVB maturation and fusion with the plasma membrane [26].

Non-enveloped viruses also hijack the ESCRT complex and secrete their virions or viral genome inside EVs, as observed with the Hepatitis A virus (HAV), Hepatitis E virus (HEV), Enterovirus 71 (EV71) and Bluetongue virus [27,28,46,47,48,49,50]. These so-called quasi-enveloped viruses can avoid immune recognition and enhance their tropism for different cell types while exiting cells in a non-cytolytic manner. The HAV capsid recruits ESCRT-III, VPS4 and ALIX, allowing the inward budding of the viral capsid inside MVBs, and its later release within EVs [27,28]. The HEV protein pORF3 interacts with TSG101, leading to the recruitment of the viral capsid and its packaging into MVBs [46] before HEV-EV Rab27a-dependent secretion [47]. The EV71 capsid and RNA are also released inside EVs in a non-lytic manner [48]. Lastly, Bluetongue virus packaging into EVs is mediated by a nonstructural viral protein, NS3, which is necessary to interact with TSG101 [49] and allow uptake inside MVBs [50] (Figure 1B1).

Moreover, some viruses also hijack ESCRT and Rab-GTPase proteins to fulfill their replication cycle without entering MVBs. First, viruses recruit the ESCRT complex to acquire their viral envelope. For instance, the human immunodeficient virus (HIV) recruits ESCRT-I, -II,-III, ALIX and VPS4 to bud from the plasma membrane [29] (Figure 1B2). Similarly, the yellow fever virus NS3 protein mediates virus budding inside the endoplasmic reticulum (ER) by binding ALIX, which potentially recruits the ESCRT complex [51]. Second, some viruses forming their capsid inside the nucleus reach the cytoplasm by hijacking the ESCRT complex. For example, herpes simplex virus-1 (HSV-1) nuclear envelopment complex (NEC) inserted inside the inner nuclear membrane recruits ESCRT-III and binds the mature capsid. This allows the internal budding of HSV-1 capsid inside the nuclear envelope in an ALIX, TSG101 and ESCRT-II independent manner. The viral capsid is then released in the cytosol by fusion with the outer nuclear envelope. HSV-1 later acquires its final envelope by budding inside trans-Golgi vesicles and is secreted by exocytosis [30,31] (Figure 1B3). Other viruses also hijack Rab-GTPases to transport their viral material, such as the Influenza virus (IAV) with Rab10 [16], or to enhance their replication cycle, as observed with Hepatitis C virus (HCV) and Rab27a [52]. Finally, viruses can hijack the ESCRT machinery to form a viral replicase complex. For instance, Tomato bushy stunt tombusvirus p33 replication protein hijacks VPS4, ESCRT-I and ESCRT-III and mediates the sorting of the viral RNA genome inside the peroxisome, where the virus replicates and is protected from cell immune surveillance [53,54].

## 3. EV Modulation of Virus Infectivity

### 3.1. EVs from Virus-Infected Cells Help Promote Infection in Healthy Cells

EVs have a prominent role in the viral spread and the infectivity of healthy cells. Many viruses pass infectious particles or genetic material to surrounding cells through EVs. This can provide both enveloped and non-enveloped viruses a cloak that hides them from immune surveillance while allowing virus dissemination, as mentioned above for HCMV, HHV-6, SARS-CoV-2, DV, HBV, HAV, HEV, EV71 and Bluetongue virus. For instance, DV-EVs containing virus-like particles transfer the virus to non-infected mosquito cells [55]. Moreover, virus-EVs also transport viral genomes that can induce active infection of recipient cells. Sera from Human pegivirus-infected subjects contain EV-like particles, which transport pegivirus RNA genome to recipient T cells, B cells, NK cells and monocytes and mediate active replication of the viral RNA inside these cells [56] (Figure 2A).

Some viruses alter the lipid and protein composition of EVs to increase binding and uptake into new cells. For instance, HAV, enteroviruses, EBV, Zika virus [57] and HIV [58] infected cells secrete EVs enriched in PS. HAV-EVs are enriched in PS and cholesterol. While PS binds the HAV cellular receptor 1 (HAVCR1) and mediates endocytosis, cholesterol binds NCP1 endosomal protein and fuses the endosomal and the EV membranes, consequently releasing the HAV capsid and genome inside the cell [59]. Similarly, enteroviruses (poliovirus, human rhinovirus and coxsackievirus B3 (CVB3)) are packaged inside autophagosome-like organelles enriched in PS, which mediates viral entry in a virus receptor-dependent fashion [60]. During a latent infection, EBV-EVs containing viral miRNA are uptaken by recipient DCs due to PS binding to the TIM-1 receptor on DCs [61]. Similarly, EBV-infected lymphoma cells secrete EVs containing viral miRNA and PS and are integrated by monocytes [62] (Figure 2A).

The unique composition of EVs can also permit virus uptake in a receptor-independent fashion and alter viral tropism, as observed with EV71, HCV, HBV and severe fever with the thrombocytopenia syndrome (SFTS) virus. EV71-EVs containing the viral genome are integrated by recipient cells in a receptor-independent manner, as the knockdown of the viral receptor did not infer with virus uptake by recipient cells [63]. Similarly, HCV-EVs contain viral RNA and host miR-122 and Ago2 protein, both of which are host factors hijacked by HCV for its replication. These HCV-EVs induce active infection of hepatocytes in a receptor-independent manner. Indeed, the presence of antibodies targeting HCV glycoproteins E1 or E2 or the host proteins involved in HCV entry inside the cells (SB-RI, ApoE, CD81) did not impair viral infection [64]. SFTS-virus-EVs contain the viral NS proteins and the viral particle, and induce active infection of recipient HeLa cells in a viral receptor-independent manner, as suggested by the absence of the effect of anti-SFTS virus antibodies [66]. Furthermore, HBV-EVs contain HBV DNA, RNA and proteins that can be integrated by natural killer (NK) cells despite HBVs tropism for hepatic cells [65] (Figure 2A).

EVs also increase cell susceptibility to virus infection by transporting virus-targeted host receptors to cells that do not normally express them at high levels. For instance, EVs transport SARS-CoV-2 viral receptor ACE2 [67] and tetraspanin CD9 [68] between non-infected endothelial cells. CD9 facilitates ACE2 aggregation at the cell surface of recipient cells, which enhances their sensitivity to SARS-CoV-2 infection. Similarly, HIV-1 viral receptor CCR5 is transferred inside microparticles to peripheral blood mononuclear cells (PBMCs) but also endothelial cells not endogenously expressing CCR5, thus allowing endothelium infection by HIV-1 [69] (Figure 2B).

Virus-EVs containing viral RNA can also facilitate the infection of recipient cells. HIV-infected monocyte-derived macrophages secrete microvesicles and EVs containing HIV-1 RNA, potentially HIV reverse transcriptase, host cytokines (interleukin (IL) -3, -4, -8, -17, leptin, tumor necrosis factor α (TNF-α)), antigen-presenting receptors and adhesion factors (HLA, CD14, CD74, CD44R5, Fc receptor, fibronectin and galectin-3) and have specific lipid composition (PS). These HIV-1-EVs control infection of recipient T cells through CD4-independent clathrin-mediated endocytosis [58]. Narayanan et al. showed that HIV-1-EVs secreted by T cell lines contain viral TAR RNA, Gag and Gp160 proteins and host miRNA machinery proteins (Dicer and Drosha) [68]. HIV-1-EVs transfer of TAR RNA inhibits apoptosis by down-regulating Bim, thus promoting recipient Jurkat T cells infection by HIV-1 [72]. In addition, the Human T-lymphotrophic virus type 1 (HTLV-1)-EVs contain viral RNA and proteins but also host the following adherent proteins: CD45, ICAM-1 [70] and its receptor LFA-1 [71]. These HTLV-1-EVs do not actively infect recipient T cells but rather enhance cell-to-cell contacts between recipient cells. Respectively, CD45 and ICAM-I/ LFA-1 are known to support the formation of viral biofilm and virological synapses, thus facilitating viral infection [70,71]. Virus-EVs can also promote latent infection and enhance virus transmission between hosts. For instance, HSV-1-EVs contain both viral miRNAs (miR-H28 and -H29) that control HSV-1 replication [73] and downregulate the stimulator of IFN genes (STING), a cytoplasmic DNA sensor that activates the antiviral innate immune response in recipient cells [74] (Figure 2B).

Virus-EVs enhance viral particle transfer by packaging multiple viral particles. This packaging enables the transfer of multiple viral genomes and proteins to recipient cells, which improves survival against the cell antiviral immune response and increases overall infectivity [60,75]. This is observed with autophagosome-derived EVs for poliovirus, human rhinovirus and coxsackievirus B3 (CVB3) [60], as well as MVB-derived EVs for noroviruses and plasma membrane-derived EVs for rotaviruses [75] (Figure 2C).

### 3.2. Virus-EVs Enhance Viral Transmission between Hosts

The encapsulation of virus particles in EVs increases the rate of virus transmission by providing safe transport between hosts. For instance, when noroviruses and rotaviruses are protected inside EVs, they are not degraded while transiting through stool and the gastrointestinal tracts. This improves the active fecal–oral transmission of these viruses [75]. Another example is DV, which is transmitted from arthropods (*Aedes albopictus* and *Aedes aegypti* mosquitos) to mammalian hosts inside EVs. DV-infected mosquito cells secrete EVs containing both viral RNA (full-length viral genome, viral capsid mRNA) and proteins (viral glycoprotein E-protein) that promote active infection in mosquito cells, mouse cells, human skin keratinocytes and blood endothelial cells [76]. Interestingly, these DV-EVs are enriched in a mosquito glycoprotein containing a tetraspanin domain, Tsp29Fb, which is a putative ortholog of human CD63. Interaction of Tsp29Fb with DV E-protein is necessary for arthropod DV-EVs uptake by human recipient cells and active infection [76]. Zika virus-infected mosquito cells secrete EVs containing PS, viral glycoprotein E-protein and viral RNA that mediate active infection of both mosquito and mammalian cells (monkey endothelial Vero cells, human monocytes, endothelial vascular cells) [57] (Figure 2C).

### 3.3. Cells Generate EVs Due to Ancestral Retrovirus Sequences

The retrovirus infection of germline cells leads to long-term modification of the host genome, where the provirus becomes an endogenous retrovirus [77]. Endogenous retroviruses are very similar to long terminal repeat (LTR) retrotransposons, as they both encode gag-like, pro-like and pol-like genes, undergo reverse transcription and integrate into the host genome [77]. These findings suggest a potential common ancestor between these two elements. Interestingly, some LTR retrotransposons also mediate the formation of virus-like particles, allowing physiologic cell-to-cell communication.

For instance, the mammalian gene PEG10 and the neuronal gene ARC are both LTR retrotransposons encoding for GAG analogs. They self-assemble and encapsulate their own mRNA before being secreted in EVs [32,78]. The PEG10 gene encodes the capsid, nucleocapsid, protease and retro-transcriptase GAG domains. Segel et al. showed that flanking genes of interest with the untranslated regions of mPEG10 allows the transport of the mRNA of interest in EVs and transfer within the tissue, which could be used for gene therapy delivery [78]. The ARC gene encodes the capsid GAG domain, which encapsulates Arc mRNA and hijacks ALIX before being secreted in EVs. These EVs could then target neuron cells, in which Arc mRNA is actively translated. Interestingly, Arc regulates synaptic plasticity by controlling the synthesis of proteins necessary for long-term depression and potentiation. Thus, this transfer of Arc mRNA between neurons could have played a primordial role in mammalian cognitive functions [32] (Figure 1B4).

Moreover, the transfer of retroelements through EVs can enhance tumorigenesis. Tumor cells secrete EVs enriched in human endogenous retrovirus (HERV) elements such as Alu and LINE-1, which promote oncogenesis in recipient cells [79,80]. For instance, cutaneous T cell lymphoma secretes EVs containing HERV W elements encoding syncytin-1. Upon EV reception, syncytin-1 mediates membrane fusion between recipient cells and could improve the transfer of tumoral signals between cells [81].

## 4. Virus-EVs Modulate Immune Responses

In the previous section, we discussed the strategies adopted by viruses to hijack EV biosynthesis machinery to aid in their replication, egress and spread. In this section, we explore the relationship between EVs, viruses and the immune system and summarize the antiviral and host immunomodulatory signaling that is mediated by EVs in the context of viral infections (Table 1).

### 4.1. Virus-EVs Stimulate Antiviral Immune Responses

Virus-EVs containing either host or viral proteins or miRNAs can enhance all stages of the antiviral innate and adaptive immune response. Virus-EVs transport host miRNAs, molecules and proteins resulting in the activation of innate antiviral immune responses in uninfected recipient cells. For instance, in the late stages of IAV infection, when cells are undergoing apoptosis, the host Y5 non-coding RNA is degraded into miRNAs that are transported via EVs to uninfected recipient cells, where they induce IFN antiviral responses [82]. During HIV-1 infection, the second messenger cGAMP is produced in response to virus sensing inside the cytoplasm. cGAMP is then transported inside HIV-1-EVs with HIV-1 viral particles to recipient cells, where cGAMP activates the stimulator of interferon genes (STING) pathway [83]. Furthermore, virus-EVs also transport IFN signaling proteins, such as interferon-induced proteins with tetratricopeptide repeats (IFIT) and interferon-induced transmembrane protein (IFITM). IFITM is known for blocking enveloped virus entry [84] and activating the adaptive immune response [85], while IFIT inhibits viral replication [86]. For instance, DV-EVs containing IFITM3 prevent DV entry into recipient cells [87], while IAV-EVs contain both IFIT and IFITM activate the secretion of IL-6, MCP-1 and TNF by recipient cells [88]. Additionally, IAV-EVs coated with α-2,3 and 2,6 sialic acids on their surface can competitively block viral entry by binding IAV particles before they can interact with targeted cells [88] (Figure 3A).

Secondly, virus-EVs stimulate the secretion of chemokines and pro-inflammatory cytokines by recipient cells, which induce activation and recruitment of immune cells to the infection site. As mentioned above, the IAV-EVs transporting IFIT and IFITM activate the secretion of IL-6, MCP-1 and TNF [88]. Respiratory syncytial virus (RSV)-EVs contain both viral RNA and proteins and host RNAs and either induce the secretion of MCP-1, IP-10 and CCL5 by recipient monocytes or of CCL5, IP-10, TNF-α by airway epithelial cells, without leading to an active infection of these cells [99]. DV-EVs secreted by infected macrophages contain miRNAs and the viral NS3 protein and induce the secretion of MCP-1, IP-10, IL-10, TNF-α and CCL5 by endothelial cells, allowing the activation of the primary antiviral immune barrier [89]. DV-EVs activation of endothelial cells is associated with an increased expression of cell adhesion proteins (ICAM and VE-cadherin) and modification of the trans-endothelial electrical resistance, supporting a change in the endothelial barrier permeability [89]. Interestingly, infection of monocyte-derived DCs induces both activation of DCs and secretion of EVs containing a variety of host miRNAs and mRNAs that depends on the nature of the DV serotype. According to their functions, these miRNAs and mRNAs can induce an inflammatory response and apoptosis in cells integrating these DV-EVs [45] (Figure 3A).

Lastly, virus-EVs can stimulate robust adaptive immune responses. As mentioned before, IAV-EVs and DV-EVs transporting IFITM could mediate this activation [85,87,88]. Moreover, IAV-EVs containing MHC-I and -II proteins, and numerous viral proteins can act as a source of antigens used by DCs to initiate the adaptive immune response. However, these IAV-EVs do not activate T cells directly [88] (Figure 3A).

### 4.2. EVs Secreted by Non-Infected Immune Cells Activate the Immune System

The antiviral immune response is also activated by EVs secreted by non-infected immune cells. EVs can protect the non-infected recipient cells from viral infection. For instance, during HBV infection, activated macrophages secrete IFN-α containing EVs, which are integrated by non-infected hepatocytes through TIM-1 (the host receptor for HAV), thus protecting cells against HBV infection [100]. EVs also enhance the immune response against the virus-associated tumor cells. Activated Vδ2-T cells secrete EVs enhancing the immune response against EBV-associated tumors. These EVs contain Fas ligand, TRAIL, NKG2D, CD80/CD86 immunostimulatory ligands and MHC-I and –II. Fas ligand and TRAIL induce cell death in recipient cells, while transportation of NKG2D to recipient cells activates NK cells. The integration of CD80/CD86 and MHC complexes into the membrane of recipient cells also enhances tumoral and viral antigen presentation by EBV-infected tumor cells, resulting in activation of CD4 and CD8 specific T cell responses [101] (Figure 3A).

### 4.3. Virus-EVs Inhibit Antiviral Immune Response

Similar to how viruses adapted the means to hijack host machinery and enhance their own replication, viruses have also found ways to suppress the antiviral immune responses by using EVs to transport host and viral immunomodulatory cargos from infected to uninfected cells. Some virus-EVs contain host miRNAs that inhibit the interferon (IFN) signaling in recipient cells, making them permissive to viral infection. EV71-EVs are enriched in host miR-146a, which represses the expression of signal transducer and activator of transcription 1 (STAT1), TNF receptor-associated factor 6 (TRAF6) and Interleukin 1 receptor-associated kinase 1 (IRAK1) and eventually suppresses the IFN-1 response in the recipient cells. Consequently, IFN-1-stimulated gene factors such as BST-2/tetherin are also inhibited. EV71-EVs integration by recipient cells not only enhances viral replication but also EV secretion by inhibiting the IFN antiviral immune response in the recipient cells since BST-2/tetherin is an inhibitor of Rab27a-dependent EV secretion [63]. Similarly, Newcastle disease virus (NDV)-EVs contain host miR-1184, miR-1273f and miR-198 that inhibit IFN-β expression in the recipient cells and thus enhance NDV replication [92]. Ebola virus-EVs transport the virus nucleocapsid, which impacts the IFN-1 response in the recipient cells [93] (Figure 3B).

Viruses have also developed strategies to evade the innate immune responses through the modulation of the EV biosynthesis pathways. Viruses, such as the Tomato bushy stunt tombusvirus, hijack the ESCRT machinery to form replication centers protected from immune surveillance [53,54]. Poliovirus, rhinovirus, coxsackievirus and DV have found ways to escape autophagolysosome degradation by hiding in autophagosome-derived EVs [60,90]. Similarly, some viruses can escape phagocytic degradation. For instance, HIV-1 can enter MVB after being captured by immature DCs, and then is trafficked inside EVs, instead of being degraded. These EVs are enriched in HLA-DR1, CD63, CD9 and are associated with HIV-1 particles, which induce active infection of recipient CD4 T cells through the HIV-1 GP120 receptor [95] (Figure 3B).

As noted earlier, virus-EVs are often enriched in PS, which not only improves uptake into recipient cells, but also plays a role in the modulation of the immune response [96]. PS is a negatively charged glycerophospholipid that is asymmetrically distributed in the plasma membrane [96]. While healthy cells present PS in the inner plasma membrane leaflet, PS translocation by Floppase to the outer leaflet acts as a cell signal. For instance, PS is externalized during cell stress and apoptosis, which leads to efferocytosis of the cell by macrophages, an alternative to phagocytosis that limits inflammation [96]. These observations suggest that the transfer of PS via virus-EVs could induce efferocytosis of recipient cells while limiting the production of pro-inflammatory molecules. However, it remains to be established whether this effect benefits viral infection by limiting the antiviral immune response or benefits the host by limiting over-inflammation.

Virus-EVs also transport viral proteins that lower immune recognition through antigen presentation. For instance, HSV-1-EVs contain the viral envelope glycoprotein B (gB), which downregulates major histocompatibility complex class II (MHC-II) expression and promotes MHC-II uptake and secretion in EVs, thereby reducing the ability of recipient cells to present viral antigens and initiate immune responses [94]. Similarly, HCMV-EVs also contain gB and gH, but also Fc-γ receptor homolog gp34. Gp34 binds the neutralizing IgG antibodies and prevents them from binding functional viral particles [23] (Figure 3B).

The transport of infectious material through EVs also provides an avenue to escape recognition by neutralizing antibodies. For example, HCV-EVs transporting viral RNA and proteins are only partially recognized by neutralizing IgG antibodies [91]. Similarly, DV-infected cells secrete autophagosome-derived EVs containing viral RNA and proteins (envelope E, nonstructural NS1, prM, membrane M), which lead to active infection of recipient cells without being impaired by neutralizing antibodies [90] (Figure 3B).

Finally, some viruses-EVs transporting viral proteins permit active infection or apoptosis of immune cells. As mentioned previously, HBV-EVs lead to an active infection of the recipient NK cells despite the tropism of HBV for hepatic cells, which suppresses the antiviral immune response and promotes chronic infection [65]. Moreover, Ebola virus-infected cells secrete EVs containing the viral matrix protein VP40, which specifically induces apoptosis of recipient T cells and monocytes, potentially by inhibiting the expression of proteins of the miRNA machinery (Dicer, Drosha, Ago1) [97,98] (Figure 3B). Ultimately, virus-EVs enhance viral spreading by inhibiting a variety of innate and adaptive immune responses. These interactions between viruses and EVs not only disturb the anti-viral immune response but also lead to a wider impact on the tissue, such as virus-associated chronic inflammation, tumorigenesis and liver disease.

## 5. EV-Mediated Progression of Virus-Associated Disorders

### 5.1. Virus-EVs Induced Inflammatory Disease

The interactions between viruses, EVs and the immune system can sometimes result in overstimulation and lead to chronic inflammatory disease. Zika virus-EVs secreted by mosquito cells that contain viral RNA, and the E-protein are integrated by naïve human endothelial vascular cells resulting in a pro-inflammatory and pro-coagulant cellular state [57]. These Zika virus-EVs induce the expression of protease-activated receptors, which activate MAPKs p38, ERK1/2 and NFκB inflammatory pathways and promote the expression of pro-inflammatory cytokines. Zika virus-Evs also increase the expression of the tissue factor receptor, which is a pro-coagulant factor. These changes support the systemic inflammation observed during Zika virus infection [57]. Zika virus-Evs can also induce the differentiation of naïve human monocytes into a pro-inflammatory intermediate/non-classical phenotype and activation of TNF-α mRNA expression [57] (Figure 4A).

Some virus-Evs transmit pathogen-associated molecular patterns (PAMPS), which activate PRRs in immune cells and promote the secretion of pro-inflammatory cytokines, eventually leading to chronic inflammation, as observed for HIV-1, Epstein–Barr virus (EBV), Ebola virus and HTLV-1. HIV-Evs containing TAR RNA and miRNA not only favor T cell infection, as mentioned above, but also support chronic inflammation [72,102]. Bernard et al. showed that the serum of HIV-1 infected patients HIV-EVs contained two other viral miRNAs (vmiR88 and vmiR99) [103], which are recognized by Toll-like receptors (TLR) -3 (TAR RNA [102]), TLR-7 (TAR miRNA [102]) or TLR-8 (TAR miRNA, vmiR88 and vmiR99 [103]) in the recipient macrophages. These interactions activate the NFκB pathway and lead to the secretion of pro-inflammatory cytokines (IL-6 [102], TNF-α [102,103]) by the recipient macrophages [102,103]. Similarly, EBV encodes 49 mature miRNAs that can all be secreted inside EVs [104]. During the latent infection, EBV-EVs containing EBER1 miRNA and PS are recognized by the TIM-1 receptor of DCs. EBER1 uncapped 5′ triphosphate terminus is recognized by PRRs, leading to the activation of the antiviral immune responses and the consequential chronic inflammatory disease in individuals suffering from autoimmune disease [61]. In addition, Ebola virus-EVs containing the viral glycoprotein can promote the secretion of pro-inflammatory cytokines by the recipient monocytes, macrophages and DCs through its interaction with TLR-4 [93]. HTLV-1 infected cells also secrete EVs containing viral tax protein and mRNA and host proinflammatory molecules (GM-CSF, IL-6). Upon reception of tax, recipient DCs secrete IL-10, IL-12, IL-17A, IFN-γ and G-CSF, which could activate Th1, Th17 and cytotoxic T cells. Altogether, these HTLV-1-EVs could either enhance the antiviral immune response or support HTLV-1-associated chronic inflammation and myelopathy/tropical spastic paraparesis neurological disorder [105].

Virus-EV-induced activation of PRRs in immune cells can also be achieved indirectly. HIV-EVs containing the Nef protein downregulates the ATP-binding cassette transporter type A1 (ABCA1) in recipient macrophages, leading to a cascade of cellular modifications. Downregulation of ABCA1 reduces cholesterol efflux, resulting in the inactivation of Cdc42, decreased actin polymerization and increased abundance of lipid rafts. This influences TLR-4 concentration in lipid rafts, potentiating ERK1/2 signaling and activating NLRP3 inflammasome and interleukin-1β (IL-1β) responses [106] (Figure 4A).

Finally, uninfected cells can also secrete EVs that promote chronic inflammation. During DV infection, IL-1β is secreted as part of the antiviral immune response, which promotes the secretion of EVs by non-infected platelets. These EVs containing various host proteins are detected by macrophages and neutrophils through CLEC5A and TLR-2, respectively, resulting in the secretion of inflammatory cytokines and activation of the neutrophile extracellular trap (NET), which lead to systemic inflammation, tissue damage, and vascular permeability [107] (Figure 4A).

HBV and HCV-EVs secreted by infected hepatocytes are integrated by hepatic stellate cells (HSCs), which cannot usually be infected by HBV and HCV, thus exasperating the disease to other cell types. HBV-infected cells express the oncogenic viral protein HBx, which induces EV biogenesis by interacting with the cellular CD9, CD81 and neutral sphingomyelinase 2 (N-SMase). The HBx protein and mRNA are secreted in HBV-EVs, then integrated by HSCs in which HBx stimulates cell proliferation, supporting HBV-associated liver disease [108]. Likewise, HCV-EVs contain the upregulated host miR-19a RNA, which activates the SOC3-STAT3-TGF-β pathway in HSCs, leading to fibrosis and worsened liver disease [109] (Figure 4B).

### 5.2. Virus-EVs Promote Tumorigenesis

Such viruses as the human papillomavirus (HPV), EBV, HIV-1 and HTLV-1 are widely known for their ability to cause cancer. Not surprisingly, EVs produced from cells infected with these viruses can also have oncogenic properties. Virus-EVs can aid in the development of tumors by transferring viral oncogenes or RNAs. For instance, HPV type 16 (HPV-16) infection increases the expression of more than 50 host microRNA (miRNA) in infected cells, ten of which are upregulated and selectively packaged into EVs as a result of the E6/E7 oncogene. These miRNAs inhibit apoptosis and senescence while inducing proliferation in both the infected and recipient cells [110,111]. In addition, keratinocytes transduced with E6 and E7 oncogenes secrete EVs containing E6 and E7 mRNA, resulting in the expression of these oncogenes in nearby cells [112]. The HTLV-1 infected cells secrete EVs containing viral tax, HBZ and Env mRNAs and the viral oncogenic tax protein. These HTLV-1-EVs enhance the survival of recipient PBMCs by protecting them from FAS-mediated apoptosis due to tax-mediated up-regulation of pro-survival signaling molecules (AKT, Rb) and activation of the NFkB pathway. This supports HTLV-1-associated adult T cell leukemia/lymphoma [105] (Figure 4C).

Virus-EVs can also cause cancer cells to adopt a more aggressive and invasive phenotype. For example, the EBV-infected cells express the viral latent membrane protein-1 (LMP-1), a constitutively active signaling protein mimicking CD40. LMP-1 can induce EV secretion through a CD63-dependent mechanism [116]. In Burkitt’s lymphoma, the EBV-infected B cells secrete EVs containing LMP-1 that induce recipient B cell proliferation and differentiation into a plasma cell-like phenotype, causing a high production of IgG1 that promotes autoimmune disorders [114]. In nasopharyngeal carcinoma, LMP-1 increases the packaging of hypoxia-inducible factor 1α (HIFα) into EVs [113] and the secretion of EVs through syndecan-2 (SDC2) and synaptotagmin-like-4 (SYTL4)—NFκB pathway [117]. In recipient nasopharyngeal carcinoma cells, HIFα serves as a transcription factor that modifies E- and N-cadherin expression, resulting in an epithelial-mesenchymal transition of cancer cells that promotes cancer invasion [113]. The HIV-1-EVs secreted by T cells contain miR-155-5p that promotes the expression of proinflammatory factors (IL-6, IL-8, TGF-β) and migration molecules (collagen type I, matrix metallopeptidase 2) in recipient cervical cancer cells. In this tumoral context, IL-6 induces cancer cell proliferation and inhibits apoptosis through STAT3. Moreover, miR-155-5p targets AT-rich interactive domain (ARID2) DNA binding protein, which inhibits the migration of cervical cancer cells through the NFkB pathway. Altogether, these HIV-1-EVs promote cervical cancer proliferation and invasion [115] (Figure 4C).

Cancer escape from immune surveillance can also be facilitated by the transportation of viral regulatory factors in virus-EVs. HPV-16-infected keratinocytes expressing E7 oncogene secrete plasma-membrane-derived EVs that inhibit the adaptive immune response. These HPV-16-EVs inhibit CD40 expression and IL-12 p40 subunit secretion by recipient Langerhans DCs in the epidermis, which reduces antigen presentation and activation of anti-HPV cytotoxic T cells [118]. The Lymphoma EBV-infected cells secrete EVs containing miR-BARTs and PS, which are integrated by monocytes. miR-BARTs enhance the upregulation of IL-10, TNF-α and Arginase 1 in active M1 monocytes, which support their phenotype switch into a regulatory M2-like phenotype and promote tumor growth and lymphoma severity [62]. Similarly, gastric carcinoma EBV-infected cells secrete EVs targeting DCs and suppress their maturation, resulting in a worse prognosis for patients [119] (Figure 4C).

## 6. EVs and Viruses as Therapeutic Tools

Ultimately, the relationships between EVs and viruses play an integral role in the spread and progression of multiple pathologies. A key contributor to these pathologies is the immune system. Virus-EVs act as potent immunomodulatory molecules that can serve to either activate or suppress the immune system. This can consequently lead to an onset of chronic inflammatory diseases or permit immune evasion and escape of other pathologies, such as cancer. However, the alternative can also be true. Virus-EV interactions can be a powerful immunotherapy strategy to target a variety of disorders.

Both viruses and EVs have a wide range of therapeutic applications. EVs have reached clinical trials for a wide variety of different illnesses—from cancer [120,121,122,123] to cardiovascular disease, type 1 diabetes, neurodegenerative diseases (Huntington’s disease, Parkinson’s disease) [124], autoimmune and inflammatory diseases [125], and even for acute respiratory distress syndrome (ARDS) from SARS-CoV-2 infection [126]. Similarly, viral vectors have been used for vaccine development for decades and are now being readily explored for gene therapies, vaccines and oncolytic viral therapies [127,128,129,130]. However, EV and virus therapies both have inherent strengths and weaknesses. EVs are safe, stable, have low immunogenicity and can intrinsically cross tissue and cellular barriers but are challenging to produce and load efficiently with drugs [120,124]. In contrast, viruses are easy to manufacture and can be engineered to overexpress therapeutic payloads but have an increased risk of toxicity and are more readily cleared by the immune system [131].

In this respect, combinational virus-EV therapeutics have many promising characteristics that overcome the weaknesses of their individual counterparts. As we have discussed previously, EVs can promote the cell-to-cell spread of viral particles and genomes, permit infection of cells with low expression of the viral receptor and provide protection against immune responses. Hence, the viral therapies that spread via EVs, either naturally or by design, have the potential of reaching more target cells without clearance from the immune system. Viruses can also help overcome the low loading of therapeutic payloads in EVs by propagating in recipient cells and delivering prolonged expression of therapeutic transgenes. Virus infection is also known to promote EV production and secretion in many cells, so there is added potential for using engineered viruses with selectivity for different tissues to generate EVs with therapeutic payloads. In the following sections, we will discuss some of the recent developments using virus-EV strategies to treat different diseases (Table 2).

### 6.1. Engineering viruses to Target EVs

Adeno-associated virus (AAV) is a powerful tool for gene delivery that has been widely studied in clinical trials for therapies targeting the brain, spinal cord, liver and muscle [142]. Despite having an excellent safety profile and high efficiency in tissue transduction, there are still some drawbacks to using AAV for gene therapy. Many patients have already acquired an immunity against AAV [143], leading to the cleansing of viral particles by neutralizing antibodies and killing of the AAV-transduced cell by cytotoxic T cells [144]. As therapeutic AAVs rarely integrate into the host genome except at specific sites, lack of integration can lead to loss of gene expression after cell division, which occurs in some tissues such as the liver. Off-target gene delivery may also occur, particularly in the liver, which can lead to other side effects [144].

To overcome these issues, AAVs can be targeted to EVs to provide protection against neutralizing antibodies while keeping the ability to target specific tissues and limiting off-target infection. Maguire et al. showed that the cells transfected with DNA for producing AAV particles also secreted EVs containing the fully functional AAV (12,2% of EVs contained AAV1) [145]. Transfection of the AAV producer cells with vesicular stomatitis virus glycoprotein G (VSV-G) further increased the production of AAV-EVs. These AAV-EVs also contained VSV-G at their surface, which led to higher transduction efficiency (AAV1 and 2) [145] (Figure 5A1).

AAV-EVs have a high potential for the treatment of neurological disorders. AAV-EVs (AAV1, 2, 9) avoid immune neutralization by host antibodies in vitro and in vivo [132] and are more efficient at transducing targeted brain cells in mice [132,133]. The selectivity of AAV can also be improved further by the creation of chimeric receptors that target neurons. Gyorgy et al. fused the rabies virus glycoprotein (RVG) to the transmembrane domain of platelet-derived growth factor (PDGF-TD) and transfected this construct along with AAV producer vectors. Due to this procedure, they generated AAV-EVs expressing RVG, which binds to cells expressing α-7 nicotinic acetylcholine receptor, and enhanced AAV-EVs uptake into the brain by neurons, astrocytes and endothelial cells [132] (Figure 5A2). It has also been suggested that integrating AAV into EVs does not increase infection efficiency but rather enhances the transport of AAV from the vessels to the tissue while protecting the virus from neutralizing antibodies [146]. This study was based on in vitro observations of the potential transduction of primary astrocytes and neuronal N2A cells after incubation with AAV-EVs, AAV or AAV in suspension with EVs [146] (Figure 5A).

AAV-EVs also showed promising results against hemophilia B genetic disease. Meliani et al. transfected HEK293 cells with AAV8 plasmids encoding the human coagulation factor IX and later harvested AAV-EVs presenting TSG101 and CD9 specific markers. After intravenous injection, these AAV-EVs are protected from antibody recognition and successfully target and transduce liver cells, brain cells and skeletal muscle cells in mice. AAV-EVs could thus improve gene therapy safety and efficiency in liver genetic disease [134]. Finally, AAV-EV strategies have also been studied for cancer therapy. Transfected HEK293 cells with AAV6 plasmids expressing luciferase have been tested for the effect of AAV-EVs on non-small cell lung cancer (carcinoma and adenocarcinoma) and small cell lung cancer cell lines. These AAV-EVs were more effective at transducing cancer cell lines in comparison to AAV alone in vitro. In vivo, AAV-EVs evade neutralizing antibodies and efficiently transduce lung cancer cells after an intratumoral injection [135] (Figure 5A).

### 6.2. Natural Transport of OVs in EVs Enhances Oncolytic Viral Therapy

The intertwined relationship between EVs, viruses and activation of the immune response has a promising role in the development of oncolytic viral therapy. Oncolytic viruses (OVs) selectively infect tumor cells causing cell death, the release of tumor and viral antigens and activation of antiviral and anti-tumoral immune responses. Moreover, OVs can be genetically modified to express factors increasing the immune response, such as immune checkpoint inhibitors (anti-PD1, anti-PD-L1 and anti-CTLA-4), cytokines (IL-2, IL-12, etc.) or T cell engagers [150,151]. Many OV therapies are currently in clinical trials, with various degrees of success. For example, talimogene laherparepvec (T-VEC) is a genetically modified HSV-1 encoding granulocyte-macrophage colony-stimulating factor (GM-CSF) that was approved by the U.S Food and Drug Administration and the European Medicine Agency in 2015 for the treatment of advanced melanoma [150,152]. The triple mutant HSV-1 G47Δ OV (Delytact) has also received approval in Japan for the treatment of malignant glioma and in clinical trials for the treatment of prostate cancer, malignant pleural mesothelioma and recurrent olfactory neuroblastoma [153,154,155].

There are many OVs, such as VSV [141], HSV-1 [73] and NDV [92], which induce the secretion of EVs to improve their spread or sensitize cancer cells to infection. In addition, tumor cells secrete high levels of EVs to support tumorigenesis [14]. These EVs create an immunosuppressive environment for the tumor to grow or transfer oncogenes to healthy cells to promote tumor invasion. EVs can also promote cancer metastasis by enhancing cell migration, modulating the extracellular matrix and modifying stromal cells to prepare a pre-metastatic niche [14]. Because of these intense EVs exchanges within the tumor and the capacity of OVs to induce EV secretion to either enhance viral spreading or increase the antiviral immune response, it is interesting to study whether EVs enhance virotherapy efficiency.

Several studies have shown that EVs increase the spreading of oncolytic adenoviruses, enhance the immune response, and help target metastatic niches. Indeed, Ad5/3-D24-GMCSF leads to the secretion of EVs containing viral proteins and genomes that actively infect the recipient tumor cells, thus enhancing viral spreading [147] (Figure 5B1). Moreover, melanoma cells secrete EVs during LOAd-CD40L or LOAd-4-1BBL Ad infection that activate DCs [148] (Figure 5B2). EVs secreted by the OBP-301 Ad-infected tumor cells can reach metastatic niches. OBP-301 is an oncolytic telomerase-specific Ad studied in a phase I clinical trial. OBP-301 administration in HCT116 colorectal primary tumors in a mouse model leads to the secretion of EVs having a strong tropism toward tumor cells, which were then transported to metastatic tumors where they induce the recruitment of immune cells, just like the OV [149] (Figure 5B3). Altogether, these observations support the potential of EVs in enhancing OV therapy efficiency. Many studies are thus turning toward EVs as a vector for OVs safe and specific transport to the tumor site.

### 6.3. Engineering of OV-EV Targeted Therapies

A major challenge with OV therapy is the presence of pre-existing anti-viral immunity. OVs are also often delivered via intratumoral injection, which limits their ability to target metastatic tumors. Whereas, intravenous injection of OVs faces issues with viral clearance in the vascular system and peripheral tissues by immune cells, neutralizing antibodies and other molecules [156]. However, OVs can be genetically modified to escape host antibodies [157] and phagocytosis [158]. In addition, there are delivery strategies for OVs, such as the use of liposomes to cloak OVs [159], the use of tumor-infiltrating T cells to which oncolytic Ad binds [160], or even the use of NK cells [161] or mesenchymal stem cells [162] that get infected by the OV and then migrate to the tumor site. In addition, EVs also have the ability to naturally uptake some OVs (OV-EVs) and improve delivery to target tissues.

Cancer cells can be used as OV-EVs producers, which will have a natural tropism for cancer cells [136] (Figure 5C1). The OV-EVs can also be loaded with chemotherapeutic agents via co-incubation, which helps increase their delivery to tumors [136] (Figure 5C2). For example, Garofalo et al. created oncolytic Ad (Ad5D24-CpG) encapsulated in EVs with the chemotherapeutic agent paclitaxel (Ad-EV-Chemo) from A549 lung carcinoma epithelial cells [136]. They found that Ad-EV-Chemo increased apoptosis of tumor cells in vitro while simultaneously raising virus replication and transduction efficiency. In nude mice harboring A549 tumors, Ad-EV-Chemo improved survival and decreased tumor growth while completely modifying the transcriptome of xenograft tumor cells [136]. They then demonstrated the efficiency of the systemic injection of the combined therapy in an immunocompetent mouse model. EVs were produced by the LLC1 murine Lewis lung carcinoma cell line. Ad-EV-Chemo induced T cells activation and infiltration inside the tumor. Interestingly, the treatment leads to a localized inflammation in the peritumoral environment, suggesting that EVs both protect OVs from alerting the immune system while ensuring the specific delivery to the tumor, making systemic injection safer [137]. In another study, Garofalo et al. produced the Ad-EVs from LLC1 cells and showed in the C57BL/6 immunocompetent mouse model that intravenous but not an intraperitoneal injection of Ad-EVs leads to the successful targeting of the treatment to the tumor and infection of tumor cells [138].

Similarly, plasma membrane-derived EVs can also carry chemotherapeutic drugs (cisplatin) [163] and oncolytic Ad (Ad5) [139] to the tumor site. Ad5-EVs secreted by A549 cancer cells are protected from antibody neutralization and deliver Ad5 inside tumor cells in a viral receptor-independent manner. Moreover, this therapy also successfully targets stem-like tumor-repopulating cells, thus preventing cancer relapse [139]. Altogether, these studies support the potential of Ad-EVs and Ad-EVs- Chemo in cancer therapy, which can target tumor cells after intravenous injection due to their protection from neutralizing antibodies, eventually inducing T cell infiltration inside the tumor and decreasing tumor growth (Figure 5C).

OV-EVs can also be engineered using EV-mimetic nanovesicle drug loading technology, which allows the production of a much higher quantity of EVs containing the drug of interest. Producer cells are first transduced to express a drug and then passed step-by-step through smaller and smaller nanosized filters, eventually forcing the formation of EV-mimetic nanovesicles containing the drug [164] (Figure 5C3). Zhang et al. used this method to produce a higher amount of EVs loaded with an oncolytic Ad5-P expressing the extracellular domain of the programmed cell death protein 1 (PD-1) [140]. The team genetically modified HEK293T cells to express VSV-G transmembrane protein and then infected these cells with Ad5-P OV. Using EV-mimetic nanovesicle drug loading technology, they successfully produced a high quantity of EVs loaded with Ad5-P and presented VSV-G at their surface. They tested these Ad5-P-EVs in various cancer cell lines and in an ascitic tumor model produced by intraperitoneal injection of hepatocellular carcinoma cell line H22 in mice. Ad5-P-EVs efficiently entered the recipient cells through the interaction of VSV-G with the low-density protein (LDL) receptor. They showed that this method of transport of Ad5-P through EVs protects the virus from neutralizing antibodies, enhances virus infection and ultimately increases the production of soluble PD-1 by infected tumor cells while prolonging mice survival [140] (Figure 5C).

### 6.4. Packaging of Therapeutic miRNAs into EVs via Engineered OVs

In the past, our group has shown that it is possible to design and encode artificial miRNAs (amiRNAs) into oncolytic VSVΔ51 that get packaged into EVs and transmitted to surrounding uninfected cells [141]. This strategy allows for the use of oncolytic viruses to deliver amiRNAs that target any cellular gene product throughout the TME. In our study, we screened a library of amiRNAs to identify candidates that sensitize cancer cells to virus infection. One such amiRNA that we found targets transcripts encoded by the cellular gene ARID1A, which enhanced infection in tumors but not normal cells. ARID1A is a member of the SWI/SNF gene family encoding helicases and ATPases that regulates gene transcription by altering chromatin structure [165,166,167,168,169]. We showed that ARID1A-knockout cells display greater susceptibility to several OV platforms, including oncolytic VSVΔ51, vaccinia virus, herpes simplex virus-1 and reovirus [141]. Interestingly, two coincident discoveries by Shen et al. [170] and Pan et al. [171] showed that disruption of the SWI/SNF complex in tumor cells also leads to enhanced immunotherapy but through the use of immune checkpoint inhibitors. In addition, ARID1A has a synthetic lethal pathway with EZH2, and thus cells that lack ARID1A can be killed by the drug GSK126, a specific EZH2 inhibitor [172].

We demonstrated that EVs produced by cancer cells upon infection with VSVΔ51 encoding an amiRNA targeting ARID1A, but not a non-targeting amiRNA, can sensitize uninfected cells to GSK126 [141]. Through enhanced OV replication and synthetic lethality, we showed increased survival in mice bearing aggressive pancreatic, ovarian and melanoma tumors [141]. We also demonstrated that we can engineer an amiRNA to target PD-L1 and combine the two amiRNAs into a single virus entity for cancer therapy [141]. Taken together, these findings support the development of virally encoded, EV-delivered amiRNAs as a strategy to promote virus spread within tumors and modify the TME.

## 7. Conclusions

EV-mediated transfer of proteins, lipids or nucleic acids to recipient cells is essential in physiologic cell-to-cell communication, but also in viral infection and immune modulation. Both RNA and DNA viruses, whether they are enveloped or not, utilize the EVs pathway to secrete their viral particles, proteins and nucleic acids, but also to secrete host elements that benefit viral infection. Virus-EVs enhance viral infection by transferring viral elements to recipient cells and modulating their response toward the virus. Moreover, virus-EVs modulate many aspects of the immune system, leading to both antiviral and pro-viral responses that can drive a variety of autoimmune and chronic inflammatory diseases and cancers. EVs and viruses are both important vectors used in many therapies. In the field of oncology, EVs-based therapies and virotherapies are both being developed. Because of how virus-EVs can enhance viral spread and activate the immune response, the combination of OVs and EVs could potentiate these cancer therapies. Indeed, packaging of OV into EVs provides safe delivery and better uptake of the OV to the tumor, which enhances the overall efficacy of the virotherapy. Despite still being in development, this strategy has great potential in cancer therapy.

## Figures and Tables

**Figure 1 ijms-24-01036-f001:**
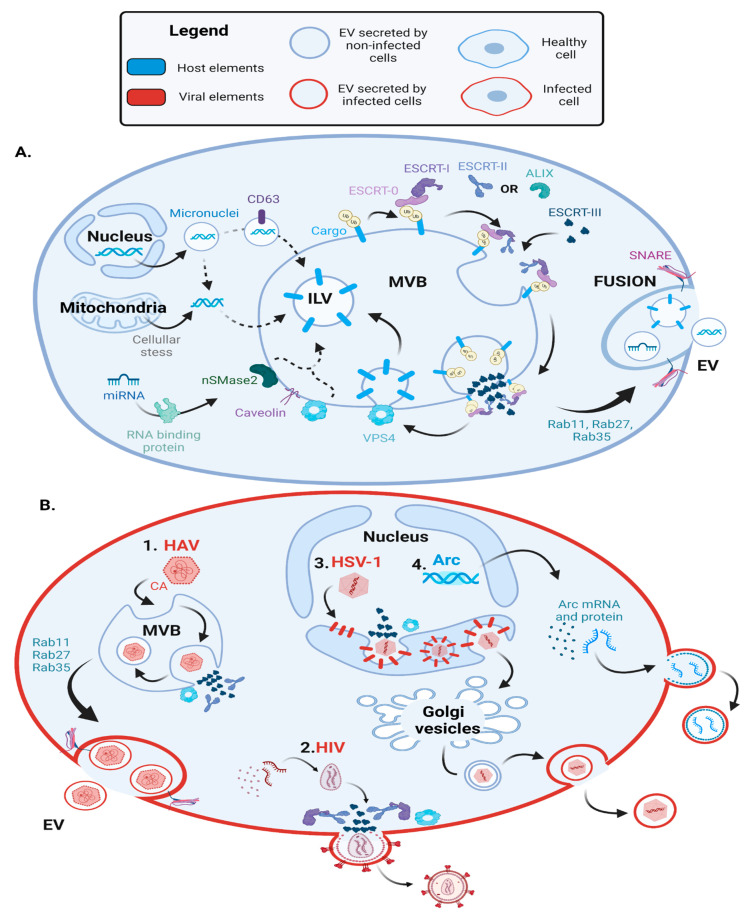
Viruses and EVs use intertwined biogenesis pathways. (**A**) EVs biosynthesis pathway: from MVBs formation to the release of EVs. EV formation requires the ESCRT complex to internalize cargos in ILVs inside MVBs, the Rab-GTPases-11, -27 and -35 to transport MVBs to the plasma membrane, and the SNARE complex for MVBs fusion with the plasma membrane and release of ILVs, becoming EVs. Ubiquitinated cargos in MVB membrane recruit ESCRT-0, which in turn recruits ESCRT-I, -II and -III, mediating membrane invagination. ESCRT-III complex forms a filament-inducing ILV modeling and scission with the help of VPS4 [20,21,22]. (**B**) Viruses hijack the EVs biosynthesis pathway. (**B1**) HAV capsid enters MVBs, is secreted in EVs and forms a “quasi-enveloped” virus. HAV viral capsid domains VP2 and VP1pX recruit ALIX, ESCRT-III and VPS4 to enter MVBs. After the fusion of the MVBs with the plasma membrane, the quasi-enveloped HAV is released. The enveloped viruses SARS-CoV-2, HCMV and HBV also enter MVBs where they acquire their viral envelope [23,24,25,26,27,28]. (**B2**) HIV acquires its envelope by hijacking the ESCRT complex. HIV envelope proteins (Env) recruit Gag and Gag/Pol polyproteins associated with the dimerized viral RNA. Gag p6 domain then recruits the ESCRT complex, ALIX and VPS4 and buds from the plasma membrane [29]. (**B3**) HSV-1 escapes the nucleus by hijacking the ESCRT complex. HSV-1 viral nuclear envelopment complex (NEC) inserted in the inner nucleus membrane recruits ESCRT-III and VPS4 and buds inside the nucleus envelope. Then, the vesicles containing HSV-1 capsid fuse with the outer nuclear membrane and release HSV-1 capsid in the cytosol. HSV-1 later acquires its viral envelope by budding inside the trans-Golgi vesicles and then by exocytosis [30,31]. (**B4**) Ancestral retrotransposon Arc protein recruits ALIX and is secreted. Arc encodes the capsid GAG domain, which self-assembles into a virus-like-capsid containing Arc mRNA. The capsid might recruit ALIX, enhancing the budding of the plasma membrane and the secretion of EVs containing Arc capsid and mRNA [32].

**Figure 2 ijms-24-01036-f002:**
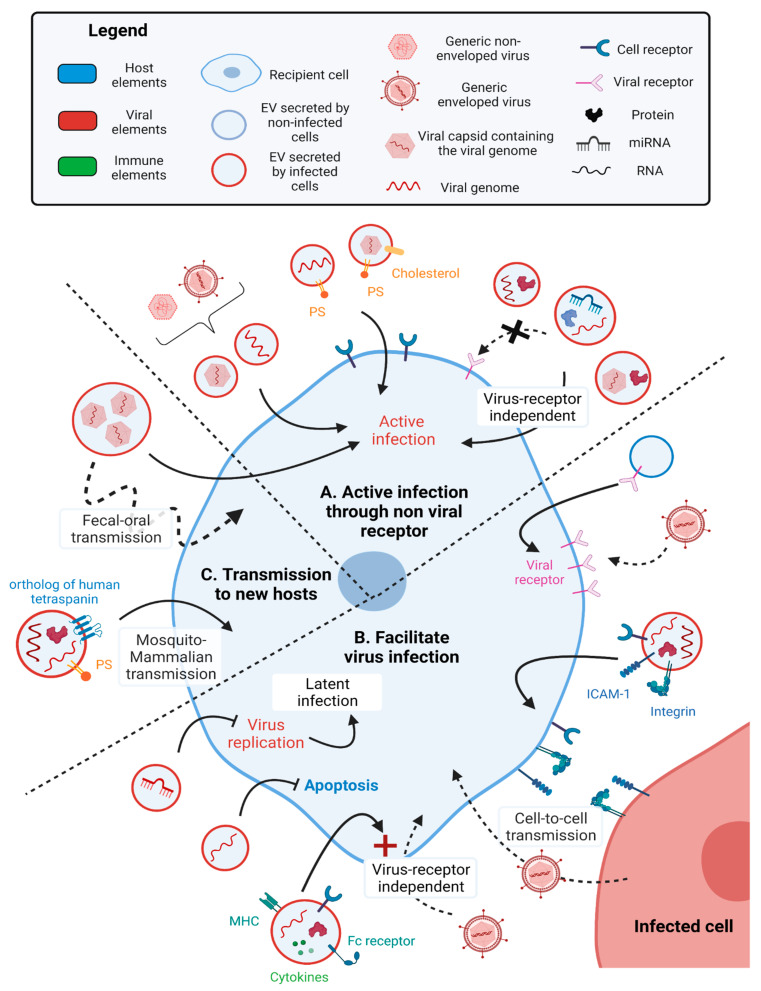
Virus-EVs increase viral uptake by recipient cells. (**A**) Virus-EVs increase viral uptake and active infection by recipient cells. Both enveloped [HCMV, HHV-6, Sars-CoV-2, DV, HBV] and non-enveloped [HAV, HEV, EV71, Bluetongue virus] viruses are secreted inside virus-EVs and induce active infection of recipient cells. The transport of the viral genome inside virus-EVs can also induce active infection of recipient cells [Pegivirus] [56]. Moreover, virus-EVs can present PS at their surface, which enhances their uptake into recipient cells [HAV [59], EBV [61,62], Zika virus [57], HIV [58], Coxsackievirus B3, rhinovirus, poliovirus [60]. Virus-EVs also induce active infection of recipient cells in a receptor-independent fashion, as observed with virus-EVs containing the viral genome and proteins, the viral capsid and proteins, or viral RNAs associated with host miRNAs and proteins [EV71 [63], HCV [64], HBV [65], SFTS virus [66]. (**B**) EV-mediated enhancement of viral infectivity. Some virus-EVs transfer the viral receptor between recipient cells, increasing viral uptake by these cells [Sars-CoV-2 [67,68], HIV [69]. Other virus-EVs containing the viral genome, viral RNAs and proteins and host adherent proteins (ICAM-1 and its receptor LFA-1, an integrin) enhance cell-to-cell transmission of the virus [HTLV-1 [70,71]. Virus-EVs containing host adherent proteins, antigen-presenting receptors, cytokines and viral RNA does not lead to infection directly but instead enhances virus transmission in a non-viral receptor fashion instead [HIV-1 [58,72]. Virus-EVs can also enhance latent infection by inhibiting virus replication [HSV-1 [73,74]. (**C**) EV mediated transmission of viruses between hosts. Single virus-EVs can transport and deliver multiple viral particles to recipient cells, thus enhancing viral infection [Poliovirus, Rhinovirus, Coxsackievirus B3, Norovirus, Rotavirus [60,75]. Some of these virus-EVs mediate virus fecal-oral transmission, resisting both the stool and the gastrointestinal tract [Norovirus, Rotavirus [75]. Virus-EVs produced by mosquitos containing the viral genome, viral RNAs and proteins, and presenting PS [Zika virus [57] or an ortholog to the human CD63 tetraspanin [DV [76] can induce active infection of mammalian cells.

**Figure 3 ijms-24-01036-f003:**
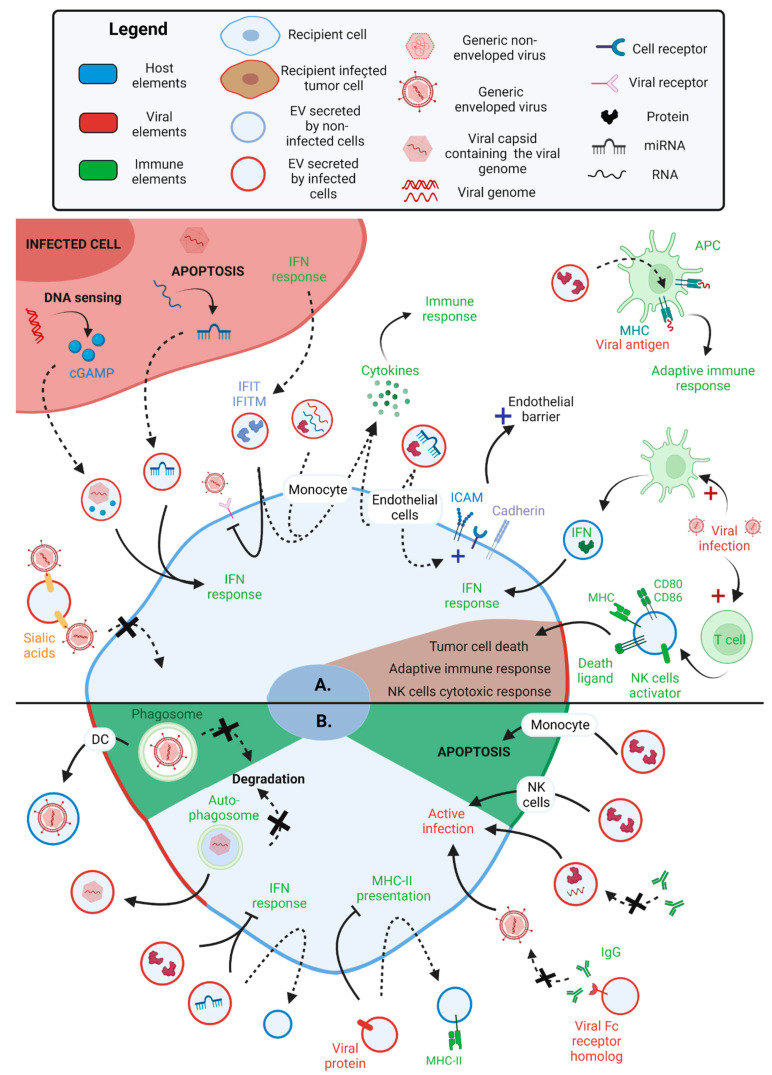
EVs and viruses modulate the immune response. (**A**) EVs secreted by both infected and non-infected cells promote the antiviral immune response. Virus-EVs enriched in transmembrane sialic acids bind viral particles and prevent them from infecting new cells [IAV [88]. Viral infection induces the production of molecules such as cGAMP due to viral DNA sensing [HIV [83] or miRNA due to virus-induced apoptosis [IAV [82]. These molecules can be transported via EVs to activate downstream antiviral IFN response in recipient cells. Virus-EVs also transfer IFN-induced molecules (IFIT and IFITM) that inhibit virus entry and enhance pro-inflammatory cytokine secretion [DV [87], IAV [88]. Virus-EVs containing various host and viral proteins and RNAs can also induce the secretion of pro-inflammatory cytokines by recipient immune cells [RSV [99] and endothelial cells [DV [89], activating the antiviral immune response. They also activate the expression of adherent proteins by endothelial cells, thus strengthening the endothelial barrier [DV [89]. Virus-EVs transporting viral proteins transporting virus antigens to antigen-presenting cells (APC) activate the adaptive immune response [IAV [88]. EVs secreted by APC activated by the viral infection transfer IFN to recipient cells [HBV [100]. EVs secreted by activated Vδ2-T cells transport immune proteins to infected tumor cells [EBV [101]. For instance, EVs transporting death ligands induce tumor cell death. EVs transferring NK cells activator enhance tumor cells death by NK cell cytotoxic response. EVs containing CD80/86 and MHC molecules increase the adaptive immune response against tumor cells [101]. (**B**) Virus-EVs enhance viral escape from the immune response. Some viruses escape degradation inside DCs phagosome and enter MVBs to be secreted in EVs [HIV-1]. Others escape degradation by bypassing the autophagolysosome of infected cells to be secreted in EVs [Poliovirus, Rhinovirus, Coxsackievirus B3, DV]. Virus-EVs containing host miRNA repress the antiviral IFN response in recipient cells, consequently enhancing viral replication [EV71 [63], NDV [92] and EV secretion [EV71 [63]. Virus-EVs containing viral proteins also inhibit the IFN pathway in recipient cells [Ebola virus [93]. Virus -EVs transporting viral proteins can inhibit MHC-II presentation and induce MHC-II secretion inside EVs by recipient cells, thus lowering potential immune recognition of infected cells [HSV-1 [94]. Virus-EVs transporting a viral Fc receptor homolog can bind neutralizing antibodies and prevent their association with viral particles [HCMV [23]. Virus-EVs also mediate the transfer of viral proteins and genome that induce active infection inside recipient cells while avoiding immune recognition by neutralizing antibodies [HCV [91], DV [90]. Virus-EVs also target immune cells by transporting viral proteins that either induce active infection [HBV [65] or apoptosis [Ebola virus [97,98] of recipient immune cells.

**Figure 4 ijms-24-01036-f004:**
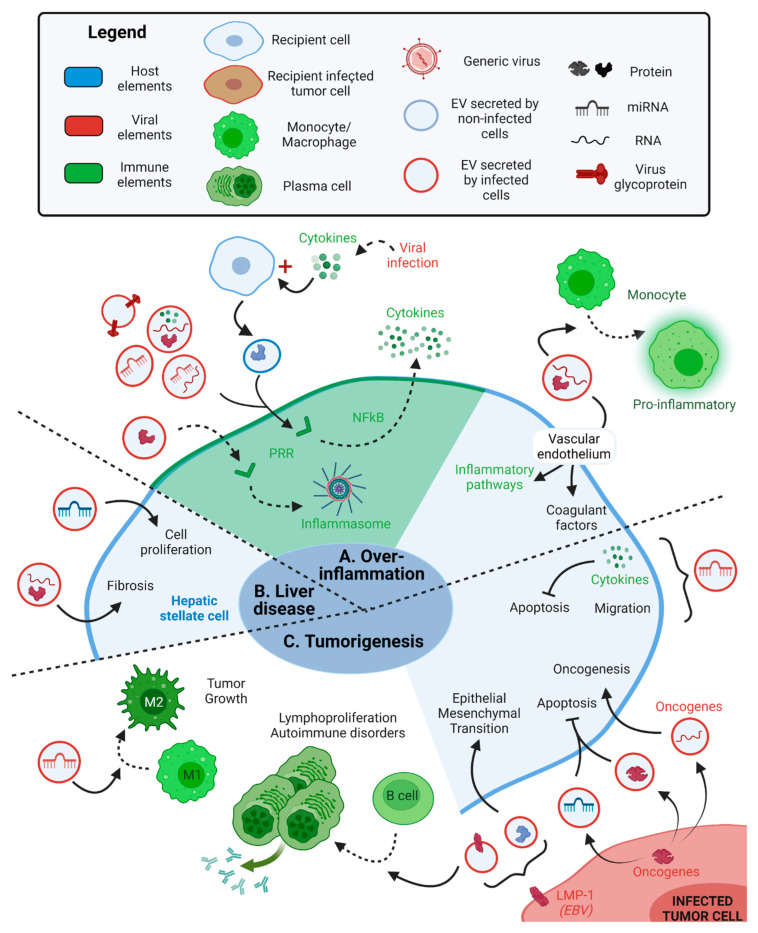
EVs mediate virus-associated disorders. (**A**) EVs secreted by both infected and non-infected cells lead to over-inflammation. Virus-EVs containing viral proteins can indirectly activate the inflammasome in immune cells, supporting over-inflammation [HIV [106]. Virus-EVs containing viral proteins, glycoproteins, miRNAs and or mRNAs can also activate the secretion of pro-inflammatory cytokines through PRR recognition in immune cells, leading to chronic inflammation [HIV-1 [72,102,103], EBV [61], Ebola virus [93], HTLV-1 [105]. Virus infection induces the release of pro-inflammatory cytokines in the environment, which activate the secretion of EVs by non-infected cells. These EVs contain host proteins that activates the secretion of pro-inflammatory cytokines by recipient immune cells through PRR recognition [DV [107]. Virus-EVs containing viral proteins and mRNAs induce the phenotype switch of recipient monocytes into a pro-inflammatory phenotype and promote inflammatory and coagulant pathways in recipient endothelial cells [Zika virus [57]. (**B**) EVs secreted by infected cells are associated with liver disease. Virus-EVs containing viral proteins and RNA or host miRNA are integrated by hepatic stellate cells and either stimulate cell proliferation or fibrosis in the liver [HBV [108], HCV [109]. (**C**) EVs secreted by infected tumor cells enhance tumorigenesis. Infected tumor cells expressing oncogenes can be associated with the secretion of virus-EVs transferring viral oncogene proteins [HTLV-1 [105] or host miRNAs [HPV-16 [110,111] to recipient cells, which inhibit apoptosis. These virus-EVs can also contain the viral oncogene mRNA, which induces oncogenesis in recipient cells [HPV-16 [112]. In EBV-infected cells, expression of LMP-1 viral protein induces secretion of EVs containing host proteins that induce the epithelial-mesenchymal transition in recipient cells and promote tumor migration [113]. EBV-EVs also transfer LMP-1 to B cells and induce their proliferation and differentiation into a plasma cell-like phenotype, causing IgG overproduction and a higher risk of autoimmune disorders [114].Virus-EVs transporting host miRNA also enhance migration and proliferation of recipient cancer cells by inducing expression of proinflammatory cytokines that limit apoptosis [HIV [115].

**Figure 5 ijms-24-01036-f005:**
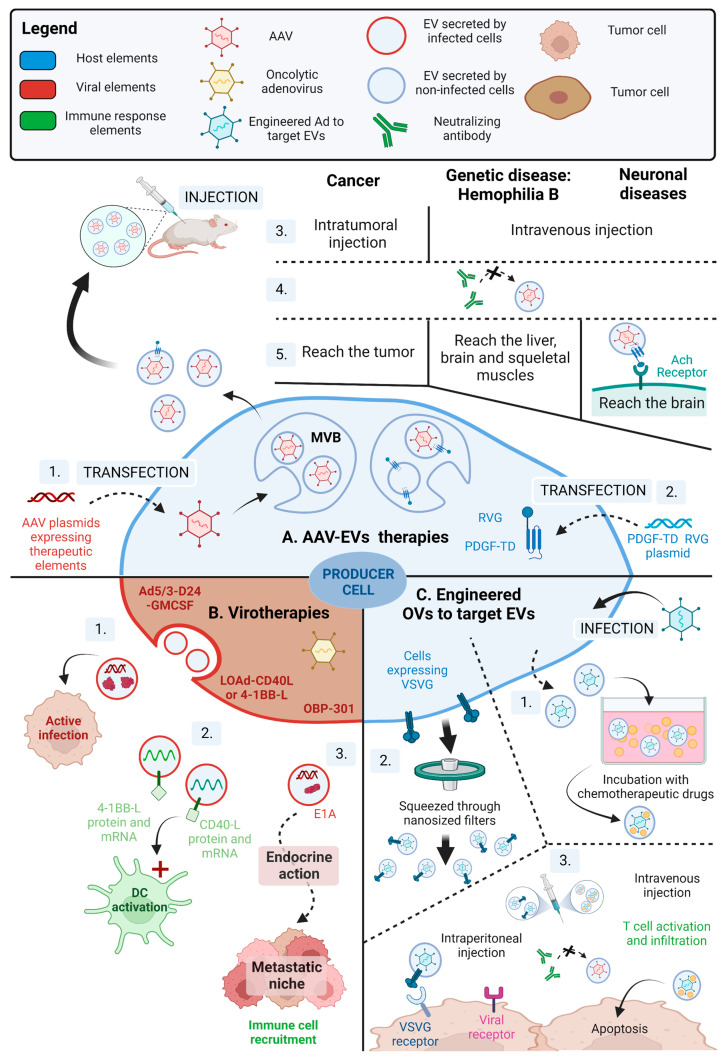
EV-guided improvement of virotherapies. (**A**) AAV-EVs-based therapies. (**A1**) Producer cells (HEK293 or 293T cells) are transfected with AAV plasmids expressing different therapeutic elements to be delivered to the appropriate tissue. (**A2**) In addition, producer cells can be co-transfected with other plasmids. For instance, PDGF-transmembrane domain fusioned to RVG allows targeting of RVG to EVs thanks to PDGT-TD. Then, RVG mediates EVs binding to the acetylcholine (Ach) receptor of neuronal cells (**A5**) [132]. (**A3**) Producer cells then naturally produce EVs, with a small proportion containing AAV particles and PDGT-TD-RVG construct. Injection of AAV-EVs into mice intratumorally or intravenously (**A3**), allows evasion of neutralizing antibodies (**A4**), increases delivery to target tissues and improves transduction efficiency (**A5**) [132,133,134,135,145,146]. (**B**) EVs improve efficacy of Ad virotherapy. PC-3 prostate and A549 lung tumor cells infected with Ad5/3-D24-GMCSF secrete EVs containing viral proteins and DNA, which actively infect recipient cells [147]. Melanoma cells (Mel526) infected with LOAd-CD40L or -4-1BB-L secrete EVs containing CD-40L or 4-1BB-L protein and mRNA respectively, which both activate DCs, thus enhancing the adaptive immune response [148]. In mice models, HCT110 primary tumor cells treated with OBP-301 Ad virotherapy secrete EVs containing the E1A viral protein and the viral DNA that can migrate to metastatic niches and induce immune cell recruitment [149]. (**C**) Engineered Ad virotherapy to target EVs. (**C1**) Producer cells are infected with the oncolytic Ad [136]. (**C2**) Producer cells can then naturally secrete EVs, with a certain fraction containing the oncolytic Ad. Incubation of Ad-EVs with a chemotherapeutic drug allows uptake of the drug inside the EVs (Ad-EV-Chemo) [136]. In vitro, Ad-EV-Chemo mediates tumor cell apoptosis and enhances cell transduction [136]. In vivo, intravenous injection mediates an increase in T cell activation and infiltration inside the tumor [137]. (**C3**) Higher quantity of Ad-EVs can be generated by passing producer cells through decreasingly smaller nanosized filters. The use of producer cells expressing VSV-G allows the formation of Ad-EVs presenting VSV-G at their surface. VSV-G then mediates binding to tumor cells through the VSV-G receptor. In vivo intraperitoneal injection of Ad-EVs in an ascitic tumor model in mice leads to both neutralizing antibody escape and successful targeting of the Ad to tumor cells in a virus receptor-independent manner [140].

**Table 1 ijms-24-01036-t001:** Summary of immune system modulation by Virus-EVs.

	Immune Mechanism	EV Mechanism	Viruses	Effect	References
Activation	Activation of Pattern Recognition receptors	Transportation of pathogen/danger associated molecular patterns (e.g., ssRNA, dsRNA, unmethylated CpG DNA)	Various (e.g., IAV, DV, RSV, etc.)	Intiation of innate and adaptive antiviral responses. Production of pro-inflammatory cytokines Protection of uninfected cells	[82]
	Activation of STING	Transport of cGAMP	HIV-1	IFN production, activation of innate and adaptive antiviral responses	[83]
	Activation of IFN signaling	Transport of IFIT/IFITM	DV, IAV	Intiation of innate and adaptive antiviral responses. Protection of uninfected cells	[84,85,86,87,88]
	Upregulation of cell adhesion proteins	EV induced activation of endothelial cells	DV	Immune cell recruitment and infitration	[89]
	Induction of Apoptosis in infected cells	Transport of miRNAs and mRNAs from infected DCs	DV	Induction of aptoposis/inflammatory responses in recipient cells	[45]
	Generation of Adaptive Immunity	Transfer of viral antigens to APCs	various (e.g., DV, IAV)	Generation of antibodies, T cell responses	[85,87,88]
Evasion	Viral Cloaking	Transport of virus particles or genomes	HCMV, HHV-6, SARS-CoV-2, DV, HBV, HAV, HEV, EV71 & Bluetongue virus	Evasion of antibodies/phagocytes	[55,56,57,58,59,60,61,62,90,91]
	Evasion of immune surveillance	Formation of immune protected replication centers with ESCRT machinery.	Tomoato bushy stunt tombusvirus	Decrease in antiviral responses	[53,54]
Inhibition	Knockdown of IFN expression	Transport of viral miRNAs that target IFN signaling genes	EV71, NDV	Decreased MHC presentation. Inhibition of antiviral responses.	[63,92,93]
	Downregulation of MHC expression	Transportation of viral proteins (e.g., gB, gH, gp34) that decrease MHC expression in recipient cells	HSV-1, HCMV	Inhibition of T cell responses	[23,94]
	Infection of immune cells	New tropisms for viruses through EV transport or evasion of phagocytosis	HBV, HIV-1	Depletion of immune cells	[65,95]
	Induction of apoptosis/efferocytosis in immune cells	Transportation of viral proteins (e.g., VP40) that induce aptoptosis; or uptake of EVs high in PS content	Ebola	Decreased immune responses	[96,97,98]

**Table 2 ijms-24-01036-t002:** Summary of Virus-EV therapies.

Virus	Target Pathology	Producer Cells	Method of Production	In Vivo Model	Evasion of Neutralizing Antibodies	Effect of Virus-EV Therapy	References
AAV 1, 2, 9	Neurological disorders	HEK293T cells	Producer cells transfected/transduced with the viral vectors. Cell debris removed by sequential centrifugation before virus-EVs are collected by ultracentrifugation.	Intravenous injection in nude mice and BALB/c mice	Yes	Enhance intake in the brain by neurons, astrocytes and endothelial cells.	[132] Figure 5A
		HEK293T cells transfected with PDGF-TD fused with RVG			Not shown		
AAV9		HEK293 cells		Intravenous injection in BALB/c mice	Not shown	Low dose of AAV-EVs allow efficient gene delivery inside the central nervous system, without toxicity.	[133]
AAV8 (encoding human coagulation factor IX)	Hemophilia B genetic disease			Intravenous injection in hemophilia B C57BL/6 mice	Yes	Low-dose of AAV8-EVs in hemophilia B mice enhances the efficacy of the gene delivery therapy, and corrects clotting deficiency.	[134] Figure 5A
AAV6	Lung cancer			Intratumoral injection in NOD SCID mice harboring subcutaneous lung (A549) tumors.	Yes	Enhance gene delivery efficiency in lung cancer.	[135] Figure 5A
Ad5D24-CpG oncolytic virus		A549 lung carcinoma epithelial cells	Similar to above, but co-incubated with Paclitaxel	Intravenous injection in BALB/c nude mice harboring subcutaneous lung (A549) tumors.	Not shown	Improve efficacy of OV and chemotherapy in the tumor by increasing specific delivery.	[136] Figure 5C
		LLC1 murine Lewis lung carcinoma cell line		Intravenous injection in C57BL/6 mice harboring lung subcutaneous (LLC1) tumors.	Yes		[137] Figure 5C
			Producer cells transfected/transduced with the viral vectors. Cell debris removed by sequential centrifugation before virus-EVs are collected by ultracentrifugation.	Intravenous or intraperitoneal injection in C57BL/6 mice harboring subcutaneous lung (LLC1) tumors.	Not shown	Intravenous injection induces a specific targeting of the treatment to the tumor.	[138] Figure 5C
Ad5 hTERTp-E1A (plasma membrane derived EV)	Cancer	A549 lung carcinoma epithelial cells		Intratumoral injection in nude mice harboring subcutaneous lung (A549) tumors. Intraperitoneal injection in BALB/c nude or C57BL/6 mice harboring intraperitoneal hepatic (H22), lung (A549) or ovarian (A2780) tumors.	Yes	Enhance cytolytic effect on tumor cells, and on tumor-repopulating cells	[139] Figure 5C
Ad5-P (encoding PD-1 extracellular domain)	Cancer	HEK293T cells expressing VSV-G.	EV-mimetic nanovesicle drug loading technology: producer cells infected with the virus and passed through sequentially smaller nanosized filters, leading to formation of virus-Evs. Virus-Evs are then collected by centrifugation using iodixanol gradient	Intraperitoneal injection in C57BL/6 mice harboring intraperitoneal hepatic (H22) tumors.	Yes	Enhance virus infection, PD-1 production and lymphocyte intratumor infiltration, while inducing a long term antitumor effect.	[140] Figure 5C
VSVΔ51 (encoding ARID1A miRNA)	Cancer	HEK293T cells	Producer cells transduced and EVs collected by ultracentrifugation	Intratumoral injection in nude CD-1 mice harboring subcutaneous pancreatic (HPAF-II) tumors and in C57BL/6 mice harboring syngeneic pancreatic (TH04) or melanoma (B16-F10) tumors.	Not shown	Increase suceptibility to OVs and sythetic lethality with GSK126 against pancreatic, ovarian and melanoma tumor models.	[141]

## Data Availability

Not applicable.

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
