# Peer review of "Extracellular Vesicles and Viruses: Two Intertwined Entities"

_ijms, 2023, doi:10.3390/ijms24021036_

Round 1
Reviewer 1 Report
The review is overall well organized and clearly written. I have only some minor suggestions for the Authors.
1) Lines 40-43: the reference number 7 is reported three times; I suggest the Authors to mention it only one time, at the end of the sentence at line 43.
2) Line 101: I think "ILV's" might be replaced with "ILVs"
3) Line 175: "EVsuptake" might be corrected to " EVs uptake"
4) Line 240: "cholesterols" might be replaced with "cholesterol"
5) I think that the titles of subsections 3.1, 3.2, 3.3 and 3.4 are repetitive and redundant. For example the 3.3 title ("Virus-EVs facilitate viral infection") and the 3.4 title ("Virus-EVs enhance viral transmission between hosts") communicate quite the same concept; I suggest the Authors to just delete one subsection and put togheter the paragraphs.
6) In section 7 the Authors introduce the therapeutic applications of viruses and EVs. I suggest the Authors to mention also the diagnostic apllication of such vectors. Among the works in the literature I suggest to cite the following:
- Ion, D. et al. An Up-to-Date Review of Natural Nanoparticles for Cancer Management. Pharmaceutics 2022, 14(1). DOI: 10.3390/pharmaceutics14010018
- Huang, T.; Deng, C.X. Current Progresses of Exosomes as Cancer Diagnostic and Prognostic Biomarkers. International Journal of Biological Sciences 2019, 15. DOI: 10.7150/ijbs.27796
- Ailuno, G. et al. Exosomes and Extracellular Vesicles as Emerging Theranostic Platforms in Cancer Research. Cells 2020, 9(12). DOI: 10.3390/cells9122569
7) Figures should be placed before the References list.
8) I think that the quality of this review might be further improved by adding a Table summarizing the works cited in section 7. The table might report the type of virus used, the target pathology for the developed viral nanocarrier, the in vitro or in vivo tests performed and so on.
Author Response
The review is overall well organized and clearly written. I have only some minor suggestions for the Authors.
We would like to thank the reviewer for taking the time to read our review and for offering suggestions for improvement.
1) Lines 40-43: the reference number 7 is reported three times; I suggest the Authors to mention it only one time, at the end of the sentence at line 43.
We have tried removing in text citations that are overly repetitive. However, we believe there are also many points that need to be supported with references and would prefer to keep them imbedded for each line to ensure readers can find that information for themselves if they would like to know more.
2) Line 101: I think "ILV's" might be replaced with "ILVs"
Corrected
3) Line 175: "EVsuptake" might be corrected to " EVs uptake"
Corrected
4) Line 240: "cholesterols" might be replaced with "cholesterol"
Corrected
5) I think that the titles of subsections 3.1, 3.2, 3.3 and 3.4 are repetitive and redundant. For example the 3.3 title ("Virus-EVs facilitate viral infection") and the 3.4 title ("Virus-EVs enhance viral transmission between hosts") communicate quite the same concept; I suggest the Authors to just delete one subsection and put togheter the paragraphs.
We apologize for the repetitiveness. We have combined several subsections in section 3 and renamed some of the subsection titles to be more descriptive of their content.
6) In section 7 the Authors introduce the therapeutic applications of viruses and EVs. I suggest the Authors to mention also the diagnostic apllication of such vectors. Among the works in the literature I suggest to cite the following:
- Ion, D. et al. An Up-to-Date Review of Natural Nanoparticles for Cancer Management. Pharmaceutics 2022, 14(1). DOI: 10.3390/pharmaceutics14010018
- Huang, T.; Deng, C.X. Current Progresses of Exosomes as Cancer Diagnostic and Prognostic Biomarkers. International Journal of Biological Sciences 2019, 15. DOI: 10.7150/ijbs.27796
- Ailuno, G. et al. Exosomes and Extracellular Vesicles as Emerging Theranostic Platforms in Cancer Research. Cells 2020, 9(12). DOI: 10.3390/cells9122569
Thank you for the suggestions. we have added the references to this section (page 12, lines 584, Refs 121-123)
7) Figures should be placed before the References list.
We have moved the figures.
8) I think that the quality of this review might be further improved by adding a Table summarizing the works cited in section 7. The table might report the type of virus used, the target pathology for the developed viral nanocarrier, the in vitro or in vivo tests performed and so on.
We have added a new table to this section (now section 6) to summarize some of the key points, as suggested.
Reviewer 2 Report
In this review article, the authors review the interaction between extracellular vesicles (EVs) and viruses. After a concise summary of EV biosynthesis, they discuss how viruses manipulate these pathways to ensure their own survival. They describe the role of EVs in virus transport and in the pathogenesis of viral diseases, while also discussing the complex relationships between EVs, viruses, and the immune system. Finally, they also provide useful insights into the therapeutic utility of viruses and EVs in combination, with a particular focus on tumors.
The article is well structured and understandable, the logic of the review is well followed, and the aspects highlighted by the authors are indeed important aspects of the relationship between EVs and viral infections.
The figures are clear and easy to understand. The literature used is relevant and up-to-date.
There are some minor typos in the text, which should be corrected. The notation 5C3 is missing in Figure 5 and should be corrected.
Nevertheless, I recommend that the article be accepted for publication after minor revisions.
Author Response
In this review article, the authors review the interaction between extracellular vesicles (EVs) and viruses. After a concise summary of EV biosynthesis, they discuss how viruses manipulate these pathways to ensure their own survival. They describe the role of EVs in virus transport and in the pathogenesis of viral diseases, while also discussing the complex relationships between EVs, viruses, and the immune system. Finally, they also provide useful insights into the therapeutic utility of viruses and EVs in combination, with a particular focus on tumors.
The article is well structured and understandable, the logic of the review is well followed, and the aspects highlighted by the authors are indeed important aspects of the relationship between EVs and viral infections.
The figures are clear and easy to understand. The literature used is relevant and up-to-date.
Thank you for the kind words and taking the time to review our manuscript.
There are some minor typos in the text, which should be corrected.
We have carefully read through the text again and corrected various typos and grammar mistakes
The notation 5C3 is missing in Figure 5 and should be corrected.
Thank you for noticing. We have corrected the notation in the figure.
Nevertheless, I recommend that the article be accepted for publication after minor revisions
Thank you for the positive recommendation.
Reviewer 3 Report
ID: ijms-2083198-peer-
Title: Extracellular Vesicles and Viruses: Two Intertwined Entities
The manuscript summary the relationships between viruses and EVs and discuss major development from the past five years in the engineering of virus-EV therapies. This article is well written and scientific sound. However, before making it into the final publication, some minor issue must to improve the present manuscript scientifically.
Specific comments are as follows;
1. Please combind “3.1. Virus-EVs enhance viral particle transfer through new receptors” and “3.2. EVs enhance viral receptor transfer to new cells” into one section.
2. Please combind “4. Cells form EVs due to ancestral retrovirus sequences” into section new “3.5”.
3. Please combind “6.1. Virus-EVs induced inflammatory disease” and “6.2. Virus-EVs enhance liver disease” into the one section.
Author Response
The manuscript summary the relationships between viruses and EVs and discuss major development from the past five years in the engineering of virus-EV therapies. This article is well written and scientific sound. However, before making it into the final publication, some minor issue must to improve the present manuscript scientifically.
Thank you for the kind words and taking the time to review our manuscript.
Specific comments are as follows;
- Please combind “3.1. Virus-EVs enhance viral particle transfer through new receptors” and “3.2. EVs enhance viral receptor transfer to new cells” into one section.
We recognized that the subsections in Section 3 were repetitive and redundant, so we combined sections 3.1-3.3 together
- Please combind “4. Cells form EVs due to ancestral retrovirus sequences” into section new “3.5”.
We have made this change as requested. It is now section 3.3
- Please combind “6.1. Virus-EVs induced inflammatory disease” and “6.2. Virus-EVs enhance liver disease” into the one section.
We have merged these two sections together (now section 5.1).
Reviewer 4 Report
General comments:
I) The abstract was too broad and was unclear what were the key findings.
II) Several sentences lacked justifications and references.
III) The part concerning EVs carrying RNA shall be deleted. Recent experimental evidence indicated that the majority of extracellular RNA is associated with supermeres rather than small EVs and exomeres (1- Qin Zhang,Dennis K. Jeppesen, James N. Higginbotham,Ramona Graves-Deal,Vincent Q. Trinh,Marisol A. Ramirez,Yoojin Sohn,Abigail C. Neininger,Nilay Taneja,Eliot T. McKinley,Hiroaki Niitsu, Zheng Cao, Rachel Evans, Sarah E. Glass, Kevin C. Ray,William H. Fissell,Salisha Hill,Kristie Lindsey Rose,Won Jae Huh,Mary Kay Washington,Gregory Daniel Ayers,Dylan T. Burnette,Shivani Sharma,Leonard H. Rome,Jeffrey L. Franklin, Youngmin A. Lee,Qi Liu,and Robert J. Coffey. Supermeres are functional extracellular nanoparticles replete with disease biomarkers and therapeutic targets. Nat Cell Biol. 2021; 23(12): 1240–1254. 2- Tosar JP, Cayota A, Witwer K. Exomeres and supermeres: Monolithic or diverse? J. Extracellular Bio. 2021; 1:e45.).
IV) There is a confusion between an enrichment of phosphatidylserine and its localization on the external leaflet of EVs, which appeared in several parts of the manuscript.
V) Several findings were taken as a face value and were embellished. The overall manuscript lacked critical analysis.
VI) The manuscript appears as a list of facts, with no clear distinction between viruses, target cells. Not all the viruses behave in the same manner, and not all the cells have the same types of function. This yields to awkward generalizations which need to be corrected.
VII) There was insufficient information on the properties of EVs. There was an insufficient discussion on the difficulties to extract a pure population of EVs. The possible variability of the properties of EVs was not discussed.
VIII) The conclusion was relatively broad, lacked justifications and references.
Minor comments:
1) Abstract, p1: Specify which mechanisms to support “They often use similar cellular machinery for entry, packaging, and secretion/egress.”
2) Abstract, p1: The sentence is a bit speculative, rephrase and add specific findings to support “Viruses can increase EV production or manipulate EVs to spread their own genetic material or proteins,”
3) Abstract, p1: The sentence is too broad. Rephrase and provide specific examples to support “while EVs can play a key role in regulating viral infections by transporting immunomodulatory molecules and viral antigens to initiate antiviral immune responses.”
4) Abstract, p1: The sentence is a bit speculative and was unclear if there are direct interactions between EVs and virus, rephrase “Ultimately, the interactions between EVs and viruses are highly interconnected, “
5) Abstract, p1: Specify which discoveries and which diseases to support “which has led to interesting discoveries in their associated roles in the progression of different diseases, as well as the new promise of combinational therapeutics.”
6) Abstract, p1: Specify, types of virus, developments and therapies to support “In this review, we summarize the relationships between viruses and EVs and discuss major developments from the past five years in the engineering of virus-EV therapies.”
7) Introduction, p1: Add references, and specify aspects of their physiology to support “Multicellular organisms rely on intercellular communication to regulate many aspects of their physiology”
8) Introduction, p1: Add references, and explanations to support “It defines environmental niches that regulate cell growth and behavior, and it is essential for collective defense against host pathogens.”
9) Introduction, p1: The sentence is too broad, add specific molecules (cytokines, chemokines, metabolites, growth factors, hormones, receptors, transcription factors, etc..) to support The majority of intercellular communication is mediated via transportation of bioactive molecules, such as proteins, nucleic acids, metabolites and lipids between cells [1,2].
10) Introduction, p1: This is not correct. Many extracellular molecules can interact via their respective receptors and are not transported and not diffused. Rephrase, and add references to support “Passage of these molecules can occur by passive diffusion or by transport via carrier molecules such as extracellular vesicles (EVs).”
11) Introduction, p1: Add references to support “EVs are cell-secreted membrane vesicles of various size, composition and origin that induce physiological changes on recipient cells through the delivery of bioactive molecules.”
12) Introduction, p1: I leave the choice to the authors to define microvesicles as “medium-sized vesicles”, and exosomes as “small-sized vesicles” to be consistent with the suggestions of ISEV.
13) Introduction, p1-2: The sentence is too broad and not correct. Delete or add more specific comments. PS is not a marker but rather its localization external or internal which may help to class the type of vesicles. Besides annexins are not trafficking proteins, they bind membrane phospholipids in a calcium-dependent manner. “Microvesicles and exosomes membrane composition share many characteristics, with the presence of tetraspanin (e.g. CD9, CD81, CD63), phosphatidylserine (PS) lipids, cell adhesion proteins (e.g. integrins) and intracellular trafficking proteins (e.g. Rab-GTPases, annexins) [7].”
14) Introduction, p2: Add references to support “Autophagosome-derived EVs are the result of a non-classical processing of the autophagosome.”
15) Introduction, p2: Add references to support “Instead of fusing with lysosomes, autophagosomes can fuse with endosomes or MVBs to form amphisomes, or instead fuse with the plasma membrane”
16) Introduction, p2: It is unclear, rephrase “In this review, we will discuss MVB-derived 51 EVs, that we will commonly refer to as EVs, while specifically focusing on EVs derived from the plasma membrane or autophagosomes.”
17) Introduction, p2: It is too broad and confusing. Specify pathogens, tumors antigens, factors, add explanations, and additional references to support “EVs can stimulate immune responses against pathogens and tumors by transporting antigens and immune-stimulating factors, maintain cellular homeostasis by excreting harmful components, such as nuclear DNA in the cytoplasm [9,10] and play a role in pregnancy, stem cell differentiation, and injury recovery [11].”
18) Introduction, p2: This is hazardous extrapolation. Add justification and references associated to each type of EV effects on cell. Specify cell origin to support “For instance, EVs secreted by adipose mesenchymal stem cells are essential in the control of cell proliferation, migration, apoptosis, but also in angiogenesis, nerve regeneration, and immune responses [12].”
19) Introduction, p2: Add justifications and references to support “While many EV-regulated signaling pathways are beneficial to cellular homeostasis and host immunity, there are also many cases where EVs propagate or exasperate pathological conditions.”
20) Introduction, p2: This is a little bit upbeat. Are you really convinced that EVs are detrimental to induce Parkinson and AD ? Rephrase and justify “EVs can transport misfolded amyloidogenic peptides associated with neuro- 62 degenerative diseases such as Parkinson’s and Alzheimer’s disease [13],”
21) Introduction, p2: Justify, rephrase and add specific references to support “aid in the role of 63 cancer progression by promoting cell proliferation, migration and angiogenesis [14] or 64 transport apoptotic and inflammatory molecules leading to cell death and other inflam- 65 matory diseases [15].”
22) Introduction, p2: Explain which type of relationship, add explanations to support “EVs also have a deeply interwoven relationship with viruses [16,17].”
23) Introduction, p2: Specify viruses, pathways to support “Viruses can exploit EV pathways to benefit all aspects of their life cycle, from entry to egress and modulation of host immune responses [16,17].”
24) Introduction, p2: Is it true that all EVs can alert the body of viral infection ? ustify, and add references to support “Conversely, EVs can also be a powerful tool for alerting the body of viral infections and stimulating antiviral responses.”
25) Introduction, p2: Justify, and add references to support “The interplay between viruses and EVs is incredibly important for the regulation of viral pathogenesis.”
26) Introduction, p2: It is unclear, justify and add references to support “Nothing demonstrates this better than the duality of immune modulation through EV signaling in infected cells.”
27) Introduction, p2: It is too broad and is hazardous extrapolation. Rephrase, specify which virus and which cells, justify to support “EVs that normally alert surrounding cells to the presence of virus infection and stimulate antiviral responses can be hijacked by some viruses to instead downregulate immune responses in neighboring cells and make them more susceptible to infection [16,17].”
28) Introduction, p2: Specify types of interactions and which virus to support “In this review, we focus on the past decade of discoveries regarding the intertwined interactions of EVs and viruses.”
29) EVs and viruses use intertwined intracellular pathways, Title, p2: Replace EVs by Extracellular vesicles in the title.
30) EVs and viruses use intertwined intracellular pathways, EV biosynthesis pathway, p2: Justify and add references to support “EVs are secreted from intraluminal vesicles (ILVs) contained in cellular multivesicu lar bodies (MVBs).”
31) EVs and viruses use intertwined intracellular pathways, EV biosynthesis pathway, p2: Justify and add references to support “The biosynthesis pathway begins with the maturation of endosomes into MVBs when membrane cargos recruit factors inducing the internal budding of the endosomal membrane and the formation of ILVs”
32) EVs and viruses use intertwined intracellular pathways, EV biosynthesis pathway, p2: Justify and add references to support “Cargo uptake is governed by different mechanisms depending on the type of cargo and their interactions with EV packaging complexes.”
33) EVs and viruses use intertwined intracellular pathways, EV biosynthesis pathway, p2-3: Specify interactions, and add reference to support “as well as specific interactions with lipids and proteins inserted inside MVBs.”
34) EVs and viruses use intertwined intracellular pathways, EV biosynthesis pathway, p3: I am not convinced if the machinery to sort endosomal sorting complexes is the same for sorting the EVs. Justify, and add references to support “The uptake of ubiquitinated proteins has been widely studied and is mediated by the endosomal sorting complexes required for 98 transport (ESCRT) machinery”
35) EVs and viruses use intertwined intracellular pathways, EV biosynthesis pathway, p3: Add references to support “Ubiquitinated proteins interact with ESCRT-0, which recruits ESCRT-I, including TSG101 subunit. ESCRT-I recruits in turn ESCRT-II and -III. Altogether, the ESCRT machinery mediates the invagination of MVBs and scission of ILV’s with the help of the vacuolar protein sorting-associated protein 4 (VPS4).”
36) EVs and viruses use intertwined intracellular pathways, EV biosynthesis pathway, p3: Add reference to support “Deubiquitylation is then required to sort proteins inside EVs (Figure 1A)”
37) EVs and viruses use intertwined intracellular pathways, EV biosynthesis pathway, p3: Specify the case, and add reference to support “In some cases, sumoylation also mediates protein uptake by the ESCRT machinery.”
38) EVs and viruses use intertwined intracellular pathways, EV biosynthesis pathway, p3: Specify non-classical sorting of proteins. Add references to support “Non-classical sorting of proteins into EVs can also be achieved through ESCRT-independent pathways.”
39) EVs and viruses use intertwined intracellular pathways, EV biosynthesis pathway, p3: Add references to support “ALIX is also recruited by the cell adaptor syntenin, which itself binds syndecan cargo inserted within the MVBs membrane.”
40) EVs and viruses use intertwined intracellular pathways, EV biosynthesis pathway, p3: Add references to support “Moreover, proteins can be packaged inside ILVs through a sphingolipid dependent, ESCRT-independent mechanism.”
41) EVs and viruses use intertwined intracellular pathways, EV biosynthesis pathway, p3: Add references to support “MVB membranes contain sphingolipids, which can be hydrolyzed into ceramide by neutral sphingomyelinase-2 (n-SMase-2).”
42) EVs and viruses use intertwined intracellular pathways, EV biosynthesis pathway, p3: Add references to support “In contrast to protein cargos, packaging of nucleic acids into EVs is less well understood but appears to depend on a variety of mechanisms.”
43) EVs and viruses use intertwined intracellular pathways, EV biosynthesis pathway, p3: Add references to support “MicroRNAs (miRNAs) are packaged into EVs according to sequence motifs that are recognized by RNA-binding proteins, such as heterogeneous nuclear ribonucleoprotein A2B1 (hnRNPA2B1), synaptotagmin binding cytoplasmic RNA-interaction protein (SYNCRIP), Argonaute2, Y-Box binding 1 protein 1 (YBX-1), MEX3C, major vault protein (MVP), and La protein.”
44) EVs and viruses use intertwined intracellular pathways, EV biosynthesis pathway, p3: Add references to support “Damaged gDNA can be transported out of the nucleus inside unstable nuclear membrane vesicles called micronuclei, which eventually release the gDNA in the cytoplasm.”
45) EVs and viruses use intertwined intracellular pathways, EV biosynthesis pathway, p3: Add references to support “Similarly, miDNA can leak in the cytoplasm during cellular stress.”
46) EVs and viruses use intertwined intracellular pathways, EV biosynthesis pathway, p3: Specify biosynthesis pathway. Add references to support “Tetraspanin small transmembrane proteins also have a significant role in the EV biosynthesis pathway.”
47) EVs and viruses use intertwined intracellular pathways, EV biosynthesis pathway, p3: Specify physiological processes. Add references to support “Tetraspanins are known for their role in mediating many physiological processes, such as cell migration, adhesion and signaling.”
48) EVs and viruses use intertwined intracellular pathways, EV biosynthesis pathway, p3: The sentence is too long. Rephrase “Once cargos are packaged into ILVs, MVBs either fuse with lysosomes to be degraded, or move to the plasma membrane where they fuse and release ILVs to the surrounding environment through processes regulated by the Soluble NSF Attachment Protein Receptor (SNARE) complex [28].”
49) EVs and viruses use intertwined intracellular pathways, EV biosynthesis pathway, p3: Justify and add references to support “These secreted ILVs are termed EVs.”
50) EVs and viruses use intertwined intracellular pathways, EV biosynthesis pathway, p3: Delete “However, the involvement of other Rabs in EVs biosynthesis pathway are still being discovered.”
51) EVs and viruses use intertwined intracellular pathways, EV biosynthesis pathway, p4: Is it true for all the EVS ? Justify and add references to support “Factors within the stroma can also have a significant role in determining EV mobility and distribution.”
52) EVs and viruses use intertwined intracellular pathways, EV biosynthesis pathway, p4: Add explanations and specific references to support “Finally, EVs are internalized by specific recipient cells through EV-associated transmembrane molecules, such as extracellular matrix proteins (ICAM-1, laminin, fibronectin), integrins, proteoglycans, lectins, glycolipids, phosphatidylserine (PS) and tetraspanins [11].”
53) EVs and viruses use intertwined intracellular pathways, EV biosynthesis pathway, p4: PS is present in most EVs. However, its localization on the external leaflet of bilayers found in specific EVs is worthwhile to be discussed to support “Finally, EVs are internalized by specific recipient cells through EV-associated transmembrane molecules, such as extracellular matrix proteins (ICAM-1, laminin, fibronectin), integrins, proteoglycans, lectins, glycolipids, phosphatidylserine (PS) and tetraspanins [11].”
54) EVs and viruses use intertwined intracellular pathways, EV biosynthesis pathway, p4: Specify cellular pathways. Justify and add explanations to support “EV binding to recipient cells can activate cellular pathways or lead to internalization through membrane fusion, clathrin-, caveolin- or lipid raft- endocytosis, but also through phagocytosis and micropinocytosis [3,11].”
55) EVs and viruses use intertwined intracellular pathways, Viruses hijack the EV biosynthesis pathway, p4: Specify viruses, which EVs, add justifications and references to support “Viruses have adapted strategies to hijack EV biosynthesis machinery to aid in all stages of their life cycles.”
56) EVs and viruses use intertwined intracellular pathways, Viruses hijack the EV biosynthesis pathway, p4: Is it occurs for all types of viruses ? Justify, and add references to support “Both RNA and DNA viruses can acquire their viral envelop by using the ESCRT complex and Rab-GTPases.”
57) EVs and viruses use intertwined intracellular pathways, Viruses hijack the EV biosynthesis pathway, p5: In the whole paragraph, the quite unusual word “hijack” was used. Its meaning is relatively broad. I suggest to reconsider the whole paragraph by replacing ”hijack” by a more specific word. Alternatively add more justifications and explanations to support the paragraph.
58) EVs and viruses use intertwined intracellular pathways, Viruses hijack the EV biosynthesis pathway, p4: Add references to support “For instance, HBV large hepatitis B surface proteins (LHBs) hijack 185 Rab5B, resulting in their transportation from the endoplasmic reticulum to the MVBs.”
59) EVs and viruses use intertwined intracellular pathways, Viruses hijack the EV biosynthesis pathway, p4: Is it true for all the viruses? Add references to support “These so-called quasi-enveloped viruses can avoid immune recognition and enhance their tropism for different cell types, while exiting cells in a non-cytolytic manner.”
60) EVs and viruses use intertwined intracellular pathways, Viruses hijack the EV biosynthesis pathway, p4: Justify and add explanations to support “Lastly, Bluetongue virus packaging into EVs is mediated by a nonstructural viral protein, NS3, which is necessary to interact with TSG101 [45] and allow uptake inside MVBs [46] (Figure 1B1).”
61) EVs and viruses use intertwined intracellular pathways, Viruses hijack the EV biosynthesis pathway, p4: Is it true for all the viruses? “Moreover, some viruses also hijack ESCRT and Rab-GTPase proteins to fulfill their replication cycle without entering MVBs.”
62) EVs and viruses use intertwined intracellular pathways, Viruses hijack the EV biosynthesis pathway, p4: Justify, add explanations and references to support “Moreover, some viruses also hijack ESCRT and Rab-GTPase proteins to fulfill their replication cycle without entering MVBs.”
63) EVs and viruses use intertwined intracellular pathways, Viruses hijack the EV biosynthesis pathway, p5: Justify, add explanations and references to support “Second, some viruses forming their capsid inside the nucleus reach the cytoplasm by hijacking the ESCRT complex.”
64) EVs and viruses use intertwined intracellular pathways, Viruses hijack the EV biosynthesis pathway, p5: Add references to support “For example, Herpes Simplex Virus-1 (HSV-1) nuclear envelopment complex (NEC) inserted inside the inner nuclear membrane recruits ESCRT-III and binds the mature capsid.”
65) EVs and viruses use intertwined intracellular pathways, Viruses hijack the EV biosynthesis pathway, p5: Add references to support “This allows the internal budding of HSV-1 capsid inside the nuclear envelop in an ALIX, TSG101 and ESCRT-II independent manner.”
66) EVs and viruses use intertwined intracellular pathways, Viruses hijack the EV biosynthesis pathway, p5: Add references to support “The viral capsid is then released in the cytosol by fusion with the outer nuclear envelop “
67) EVs and viruses use intertwined intracellular pathways, Viruses hijack the EV biosynthesis pathway, p5: Justify. Add references to support “Finally, viruses can hijack the 217 ESCRT machinery to form a viral replicase complex.”
68) EVs and viruses use intertwined intracellular pathways, Viruses hijack the EV biosynthesis pathway, p5: Add explanations to support “For instance, Tomato bushy stunt tombusvirus p33 replication protein hijacks VPS4, ESCRT-I and ESCRT-III and mediates the sorting of the viral RNA genome inside the peroxisome, where the virus replicates and is protected from cell immune surveillance [52,53].”
69) Virus-EVs promote viral infection, Title, p5: Replace EV by extracellular vesicle in the title.
70) Virus-EVs promote viral infection, p5: It is unclear what virus-EVs means. Virus do not have the machinery to produce EVs. Add explanations to support the paragraph.
71) Virus-EVs promote viral infection, p5: Specify viruses, justify, add references to support “EVs have a prominent role on viral spread and the infectivity of healthy cells.”
72) Virus-EVs promote viral infection, p5: Specify viruses, cells, justify, add references to support “Many viruses pass infectious particles or genetic material to surrounding cells through EVs.”
73) Virus-EVs promote viral infection, p5: Specify abbreviations in ”This can provide both enveloped and non-enveloped viruses a cloak that hides them from immune surveillance while allowing virus dissemination, as mentioned above for HCMV, 227 HHV-6, SARS-CoV-2, DV, HBV, HAV, HEV, EV71 and Bluetongue virus.”
74) Virus-EVs promote viral infection, p5: It is unclear what “cloak” means. Is that viruses are inserted into EVs or adsorbed on EVs ? Justify, add references to support “This can provide both enveloped and non-enveloped viruses a cloak that hides them from immune surveillance while allowing virus dissemination, as mentioned above for HCMV, HHV-6, SARS-CoV-2, DV, HBV, HAV, HEV, EV71 and Bluetongue virus.”
75) Virus-EVs promote viral infection, p5: Justify, add explanations and references to support “Moreover, virus-EVs also transport viral genome which can induce active infection of recipient cells.”
76) Virus-EVs enhance viral particle transfer through new receptors, Title, p5: Replace “EVs” by “Extracellular vesicles” and delete “new” in the title.
77) Virus-EVs enhance viral particle transfer through new receptors, Title, p5: Replace Specify viruses, justify and add references to support “Some viruses alter the lipid and protein composition of EVs to increase binding and uptake into new cells.”
78) Virus-EVs enhance viral particle transfer through new receptors, p5: It is unclear if EVs were enriched in PS or if the PS was detected on the external leaflet. Check to support “For instance, HAV, enteroviruses, EBV, Zika virus [56] and HIV 238 [57] infected cells secrete EVs enriched in PS.”
79) Virus-EVs enhance viral particle transfer through new receptors, p5: It is unclear if EVs were enriched in PS or if the PS was detected on the external leaflet, and add references to support “HAV-EVs are enriched in PS and cholesterol.”
80) Virus-EVs enhance viral particle transfer through new receptors, p5: Further justification is needed to support that there was a cholesterol enrichment in HAV-EVs to support “HAV-EVs are enriched in PS and cholesterol.”
81) Virus-EVs enhance viral particle transfer through new receptors, p5: It is unclear if EVs were enriched in PS or if the PS was detected on the external leaflet. Check “Similarly, enteroviruses (Poliovirus, human Rhinovirus and Coxsackievirus B3 (CVB3)) are packaged inside autophagosome-like organelles enriched in PS, which mediates viral entry in a virus receptor dependent fashion [59].”
82) Virus-EVs enhance viral particle transfer through new receptors, p5: Recent experimental evidence indicated that the majority of extracellular RNA is associated with supermeres rather than small EVs and exomeres (1- Qin Zhang,Dennis K. Jeppesen, James N. Higginbotham,Ramona Graves-Deal,Vincent Q. Trinh,Marisol A. Ramirez,Yoojin Sohn,Abigail C. Neininger,Nilay Taneja,Eliot T. McKinley,Hiroaki Niitsu, Zheng Cao, Rachel Evans, Sarah E. Glass, Kevin C. Ray,William H. Fissell,Salisha Hill,Kristie Lindsey Rose,Won Jae Huh,Mary Kay Washington,Gregory Daniel Ayers,Dylan T. Burnette,Shivani Sharma,Leonard H. Rome,Jeffrey L. Franklin, Youngmin A. Lee,Qi Liu,and Robert J. Coffey. Supermeres are functional extracellular nanoparticles replete with disease biomarkers and therapeutic targets. Nat Cell Biol. 2021; 23(12): 1240–1254. 2- Tosar JP, Cayota A, Witwer K. Exomeres and supermeres: Monolithic or diverse? J. Extracellular Bio. 2021; 1:e45.). Delete “Similarly, EBV infected lymphoma cells secrete EVs containing viral miRNA and PS and are integrated by monocytes [61] (Figure 2A).”
83) Virus-EVs enhance viral particle transfer through new receptors, p5: Add references to support “The unique composition of EVs can also permit virus uptake in a receptor independent fashion and alter viral tropism, as observed with EV71, HCV, HBV and severe fever with thrombocytopenia syndrome (SFTS) virus.”
84) Virus-EVs enhance viral particle transfer through new receptors, p5: Delete “Similarly, HCV-EVs 253 contain viral RNA and host miR-122 and Ago2 protein, both of which are host factors 254 hijacked by HCV for its replication.”
85) Virus-EVs enhance viral particle transfer through new receptors, p5: Justify, and add references to support “These HCV-EVs induce active infection of hepatocytes 255 in a receptor-independent manner.”
86) Virus-EVs enhance viral particle transfer through new receptors, p6: Specify NS proteins to support “SFTS-virus-EVs contain the viral NS proteins and the viral particle, and induce active infection of recipient HeLa cells in a viral receptor independent manner, as suggested by the absence of effect of anti-SFTS virus antibodies [64].”
87) Virus-EVs enhance viral particle transfer through new receptors, p6: Based on the experimental evidence that extracellular RNA is associated with supermeres (see comment 83). Delete “Furthermore, HBV-EVs contain HBV DNA, RNA and proteins and that can be integrated by natural killer (NK) cells despite HBV’s tropism for hepatic cells [65] (Figure 2A).”
88) EVs enhance viral receptor transfer to new cells, Title, p6: Replace EVs by extracellular vesicles. Delete “new”.
89) EVs enhance viral receptor transfer to new cells, p6: It is unclear. Justify and add references to support “EVs also increase cell susceptibility to virus infection by transporting virus-targeted host receptors to cells that do not normally express them at high levels.”
90) EVs enhance viral receptor transfer to new cells, p6: Add references to support “CD9 facilitates ACE2 aggregation at the cell surface of recipient cells, which enhances their sensitivity to SARS-CoV-2 infection.”
91) Virus-EVs facilitate viral infection, Title, p6: Replace EVs by extracellular vesicles.
92) Virus-EVs facilitate viral infection, p6: The whole paragraph 3.3 needs to be re-evaluated or deleted. The detection of RNA fragments is complicated by the fact that EVs are probably not the best RNA vehicles (see comment N° 83).
93) Virus-EVs facilitate viral infection, p6: Strong justification supported by solid references are needed or delete “Virus-EVs containing viral RNA can also facilitate the infection of recipient cells.”
94) Virus-EVs facilitate viral infection, p6: Delete “HIV-infected monocyte-derived macrophages secrete microvesicles and EVs containing HIV-1 RNA, potentially HIV reverse transcriptase, host cytokines (interleukin (IL) -3, -4, - 8, -17, leptin, Tumor Necrosis Factor α (TNF-α)), antigen presenting receptors and adhesion factors (HLA, CD14, CD74, CD44R5, Fc receptor, fibronectin, and galectin-3) and have specific lipid composition (PS). These HIV-1-EVs control infection of recipient T cells through CD4 independent clathrin-mediated endocytosis [57].”
95) Virus-EVs facilitate viral infection, p6: Delete “In addition, Human T-lymphotrophic virus type 1 (HTLV-1)-EVs contain viral RNA and proteins, but also host adherent proteins: CD45, ICAM-1 [70] and its receptor LFA-1 [71]. These HTLV-1-EVs do not actively infect recipient T cells, but rather enhance cell-to-cell contacts between recipient cells.”
96) Virus-EVs facilitate viral infection, p6: Justify and add references to support “Virus-EVs can also promote latent infection and enhance virus transmission between hosts.”
97) Virus-EVs facilitate viral infection, p6: Delete “For instance, HSV-1-EVs contain both viral miRNAs 293 (miR-H28 and -H29) which control HSV-1 replication [72] and downregulate the stimula- 294 tor of IFN genes (STING), a cytoplasmic DNA sensor which activate the antiviral innate 295 immune response in recipient cells [73] (Figure 2B).”
98) Virus-EVs facilitate viral infection, p6: Specify particles, justify and add references to support “Virus-EVs enhance viral particle transfer by packaging multiple viral particles”
99) Virus-EVs enhance viral transmission between hosts, Title, p6: Replace EVs by extracellular vesicles in the title.
100) EVs enhance viral transmission between hosts, p6: Avoid gratuitous sentence. Focus on the facts, justify and add references to support “Encapsulation of virus particles in EVs increases the rate of virus transmission by providing safe transport between hosts.”
101) EVs enhance viral transmission between hosts, p6: Justify and add references to support “For instance, when noroviruses and rotaviruses are protected inside EVs, they are not degraded while transiting through stool and the gastrointestinal tracts”
102) EVs enhance viral transmission between hosts, p6-7: Justify and add references to support “Another example is DV, which is transmitted from arthropods (Aedes albopictus and Aedes aegypti mosquitos) to mammalian hosts inside EVs”
103) EVs enhance viral transmission between hosts, p6-7: Delete “DV infected mosquito cells secrete EVs containing both viral RNA (full length viral genome, viral capsid mRNA) and proteins (viral glycoprotein E-protein) that promote active infection in mosquito cells, mouse cells, human skin keratinocytes and blood endothelial cells [75].”
104) EVs enhance viral transmission between hosts, p7: Add references to support “Interestingly, these DV-EVs are enriched in a mosquito glycoprotein containing a tetraspanin domain, Tsp29Fb, which is a putative ortholog of human CD63.”
105) EVs enhance viral transmission between hosts, p7 Rather than the presence of PS, check if PS was localized on the external leaflet of EVs to support “Zika- 317 virus infected mosquito cells secrete EVs containing PS, viral glycoprotein E-protein and viral RNA which mediate active infection of both mosquito and mammalian cells (monkey endothelial Vero cells, human monocytes, endothelial vascular cells) [56] (Figure 2C). ”
106) Cells form EVs due to ancestral retrovirus sequences, Title, p7: Replace EVs by extracellular vesicles in the title.
107) Cells form EVs due to ancestral retrovirus sequences, p7: Add explanations and references to support “These findings suggest a potential common ancestor between these two elements.”
108) Cells form EVs due to ancestral retrovirus sequences, p7: Add explanations and references to support “Interestingly, some LTR retrotransposons also mediate the formation of virus- 328 like particles, allowing physiologic cell to cell communication.”
109) Cells form EVs due to ancestral retrovirus sequences, p7: Add references to support “For instance, the mammalian gene PEG10 and the neuronal gene ARC are both LTR retrotransposons encoding for GAG analogs”
110) Cells form EVs due to ancestral retrovirus sequences, p7: Delete “They self-assemble and encapsulate their own mRNA before being secreted in EVs [77,78].”
111) Cells form EVs due to ancestral retrovirus sequences, p7: Add references to support “PEG10 gene encodes the capsid, nucleocapsid, protease and retro-transcriptase GAG domains.”
112) Cells form EVs due to ancestral retrovirus sequences, p7: Delete “Segel et al. showed that flanking genes of interest with the untranslated regions of mPEG10 allow transport of the mRNA of interest in EVs and transfer within the tissue, which could be used for gene therapy delivery [78].”
113) Cells form EVs due to ancestral retrovirus sequences, p7: Add explanations and references to support “Moreover, the transfer of retroelements through EVs can enhance tumorigenesis.”
114) Virus-EVs modulate immune responses, Title, p7 : Replace EVs by extracellular vesicles in the title.
115) Virus-EVs modulate immune responses, Virus-EVs stimulate antiviral immune responses, Title, p7: Replace EVs by extracellular vesicles in the title
116) Virus-EVs modulate immune responses, Virus-EVs stimulate antiviral immune responses, p7: Is it true for all the Virus-EVs? Delete miRNAs, justify, and add references to support “Virus-EVs containing either host or viral proteins or miRNAs can enhance all stages of the antiviral innate and adaptive immune response.”
117) Virus-EVs modulate immune responses, Virus-EVs stimulate antiviral immune responses, p7: Delete miRNAs, justify, and add references to support “Virus-EVs transport host miRNAs, molecules and proteins resulting in the activation of innate antiviral immune responses in uninfected recipient cells.”
118) Virus-EVs modulate immune responses, Virus-EVs stimulate antiviral immune responses, p7-8: Delete “For instance, in the late stages of IAV infection when cells are undergoing apoptosis, the host Y5 non-coding RNA is degraded into miRNAs that are transported via EVs to uninfected recipient cells where they induce IFN antiviral responses [82].”
119) Virus-EVs modulate immune responses, Virus-EVs stimulate antiviral immune responses, p8: Add references to support “ “Secondly, virus-EVs stimulate the secretion of chemokines and pro-inflammatory cyokines by recipient cells, which induce activation and recruitment of immune cells to the nfection site.”
120) Virus-EVs modulate immune responses, Virus-EVs stimulate antiviral immune responses, p8: Delete viral RNA in “Respiratory Syncytial Virus (RSV)-EVs contain both 377 viral RNA and proteins and host RNAs and either induce the secretion of MCP-1, IP-10 378 and CCL5 by recipient monocytes, or of CCL5, IP-10, TNF-α by airway epithelial cells, 379 without leading to an active infection of these cells [89].”
121) Virus-EVs modulate immune responses, Virus-EVs stimulate antiviral immune responses, p8: Delete miRNAs in “DV-EVs secreted by infected mac- 380 rophages contain miRNAs and the viral NS3 protein and induce secretion of MCP-1, IP- 381 10, IL-10, TNF-α and CCL5 by endothelial cells, allowing activation of the primary antivi- 382 ral immune barrier [90].”
122) Virus-EVs modulate immune responses, Virus-EVs stimulate antiviral immune responses, p8: Add references to support “Interestingly, infection of monocyte-derived DCs induce both activa- 386 tion of DCs and secretion of EVs containing a variety of host miRNAs and mRNAs which 387 depends on the nature of the DV serotype.”
123) Virus-EVs modulate immune responses, Virus-EVs stimulate antiviral immune responses, p8: Justify. Add references to support “Lastly, virus-EVs can stimulate robust adaptive immune responses.”
124) Virus-EVs modulate immune responses, Virus-EVs stimulate antiviral immune responses, p8: Add references to support “Moreover, IAV-EVs containing MHC-I and -II proteins, and numerous viral proteins can 393 act as a source of antigen used by DCs to initiate the adaptive immune response.”
125) Virus-EVs modulate immune responses, EVs secreted by non-infected immune cells activate the immune system, Title, p8: Replace EVs by extracellular vesicles.
126) Virus-EVs modulate immune responses, EVs secreted by non-infected immune cells activate the immune system, p8: Is it true for all the EVs? Justify and add references to support “The antiviral immune response is also activated by EVs secreted by non-infected immune cells.”
127) Virus-EVs modulate immune responses, EVs secreted by non-infected immune cells activate the immune system, p8: It is unclear how non-infected cells can trigger the EVs. Justify and add references to support “EVs can protect non-infected recipient cells from viral infection.”
128) Virus-EVs modulate immune responses, EVs secreted by non-infected immune cells activate the immune system, p8: It is unclear how are triggered and secreted EVs against virus associated tumor cells. Justify and add references to support “EVs also enhance the immune response against virus-associated tumor cells.”
129) Virus-EVs modulate immune responses, EVs secreted by non-infected immune cells activate the immune system, p8: Add references to support “These EVs contain Fas ligand, TRAIL, NKG2D, CD80/CD86 immunostimulatory ligands and MHC-I and –II.”
130) Virus-EVs modulate immune responses, EVs secreted by non-infected immune cells activate the immune system, p8: Add references to support “Fas ligand and TRAIL induce cell death in recipient cells while transportation of NKG2D to recipient cells activates NK cells”
131) Virus-EVs modulate immune responses, Virus-EVs inhibit antiviral immune response, Title, p8: Replace EVs by extracellular vesicles.
132) Virus-EVs modulate immune responses, Virus-EVs inhibit antiviral immune response, p9: Is it true for all the viruses? Justify, and add references to support “Similar to how viruses adapted the means to hijack host machinery and enhance their own replication, viruses have also found ways to suppress antiviral immune responses by using EVs to transport host and viral immunomodulatory cargos from infected to uninfected cells.”
133) Virus-EVs modulate immune responses, Virus-EVs inhibit antiviral immune response, p9: Specify viruses. Add references to support “Some virus-EVs contain host miRNAs that inhibit the interferon (IFN) 416 signaling in recipient cells, making them permissive to viral infection.”
134) Virus-EVs modulate immune responses, Virus-EVs inhibit antiviral immune response, p9: The majority of extracellular RNA is associated with supermeres rather than small EVs and exomeres (see comment 83). Delete “EV71-EVs are enriched in host miR-146a, which represses the expression of Signal transducer and activator of transcription 1 (STAT1), TNF receptor-associated factor 6 (TRAF6) and Interleukin 1 Receptor Associated Kinase 1 (IRAK1) and eventually suppresses the IFN-1 response in recipient cells. Consequently, IFN-1 stimulated gene factors such as BST-2/tetherin are also inhibited. EV71-EVs integration by recipient cells not only enhance viral replication but also EV secretion by inhibiting the IFN antiviral immune response in recipient cells, since BST-2/tetherin is an inhibitor of Rab27a-dependent EV secretion [62]. Similarly, Newcastle disease virus (NDV)-EVs contain host miR-1184, miR-1273f and miR-198 which inhibit IFN-β expression in recipient cells and thus enhance NDV replication [93]. Ebola virus-EVs transport the virus nucleocapsid, which impacts the IFN-1 response in recipient cells [94]”
135) Virus-EVs modulate immune responses, Virus-EVs inhibit antiviral immune response, p9: Is it true for all the viruses? Add references to support “Viruses have also developed strategies to evade innate immune responses through modulation of the EV biosynthesis pathways.”
136) Virus-EVs modulate immune responses, Virus-EVs inhibit antiviral immune response, p9: Add explanations, specify ways to support “Poliovirus, rhinovirus, coxsackievirus and DV have found ways to escape autophagolysosome degradation by hiding in autophagosome-derived EVs [59,95].”
137) Virus-EVs modulate immune responses, Virus-EVs inhibit antiviral immune response, p9: Specify viruses. Add references to support “Similarly, some viruses can escape phagocytic degradation. For instance, HIV-1 can enter MVB after being captured by immature DCs, and then is trafficked inside EVs, instead of being degraded.”
138) Virus-EVs modulate immune responses, Virus-EVs inhibit antiviral immune response, p9: It is the localization of PS on the external leaflet and not the enrichment in PS that can modulate the immune response to be consistent with the next sentences. Rephrase “As noted earlier virus-EVs are often enriched in PS, which not only improves uptake into recipient cells, but also plays a role in the modulation of the immune response [97].”
139) Virus-EVs modulate immune responses, Virus-EVs inhibit antiviral immune response, p9: Delete since it is not correctly explained. Besides justifications and references are needed. “These observations suggest that transfer of PS via virus-EVs could induce efferocytosis of recipient cells while limiting production of pro-inflammatory molecules. However, it remains to be established whether this effect benefits viral infection by limiting the antiviral immune response or benefits the host by limiting over-inflammation.”
140) Virus-EVs modulate immune responses, Virus-EVs inhibit antiviral immune response, p9: Is it true for all the viruses? Justify, and add references to support “Virus-EVs also transport viral proteins that lower immune recognition through anti-gen presentation.”
141) Virus-EVs modulate immune responses, Virus-EVs inhibit antiviral immune response, p9: Add references to support “The transport of infectious material through EVs also provides an avenue to escape recognition by neutralizing antibodies.”
142) Virus-EVs modulate immune responses, Virus-EVs inhibit antiviral immune response, p9: Specify viruses. Add references to support “Finally, some viruses-EVs transporting viral proteins permit active infection or apoptosis of immune cells.”
143) Virus-EVs modulate immune responses, Virus-EVs inhibit antiviral immune response, p10: Is it true for all the viruses? Justify. Add references to support “Ultimately, virus-EVs enhance viral spreading by inhibiting a variety of innate and adaptive immune responses. These interactions between viruses and EVs not only disturb the anti-viral immune response, but also lead to wider impact on the tissue, such as virus-associated chronic inflammation, tumorigenesis and liver disease.”
144) EV-mediated progression of virus-associated disorders, Virus-EVs induced inflammatory disease Title, p10: Replace EV by extracellular vesicles in the title.
145) EV-mediated progression of virus-associated disorders, Virus-EVs induced inflammatory disease, p10: Specify interactions. Justify, add references to support “The interactions between viruses, EVs and the immune system can sometimes result in overstimulation and lead to chronic inflammatory disease.”
146) EV-mediated progression of virus-associated disorders, Virus-EVs induced inflammatory disease, p10: See comment 83. Delete “Zika virus-EVs secreted by mosquito cells that contain viral RNA and the E-protein are integrated by naïve human endothelial vascular cells resulting in a pro-inflammatory and pro-coagulant cellular state [56]. These Zika virus-EVs induce the expression of protease-activated receptors, which activate MAPKs p38, ERK1/2 and NFκB inflammatory pathways and promote expression of pro-inflammatory cytokines. Zika virus-EVs also increase the expression of the tissue factor receptor, which is a pro-coagulant factor. These changes support the systemic inflammation observed during Zika virus infection [56]. Zika virus-EVs can also induce differentiation of naïve human monocytes into a pro-inflammatory intermediate/non classical phenotype and activation of TNF-α mRNA expression [56]”
147) EV-mediated progression of virus-associated disorders, Virus-EVs induced inflammatory disease, p10: It was unclear if cytokines were found in EVs. Justify, and add references to support “Some virus-EVs transmit pathogen associated molecular patterns (PAMPS), which activate PRRs in immune cells and promote secretion of pro-inflammatory cytokines, eventually leading to chronic-inflammation, as observed for HIV-1, Epstein-Barr virus EBV), Ebola virus and HTLV-1.”
148) EV-mediated progression of virus-associated disorders, Virus-EVs induced inflammatory disease, p10: See comment 83. Delete “HIV-EVs containing TAR RNA and miRNA not only favor T cell infection as mentioned above, but also support chronic inflammation [69,102]. Bernard et al. showed that the serum of HIV-1 infected patients HIV-EVs contained two other viral miRNA (vmiR88 and vmiR99) [103], which are recognized by Toll Like Receptors (TLR) -3 (TAR RNA [102]), TLR-7 (TAR miRNA [102]) or TLR-8 (TAR miRNA, vmiR88 and vmiR99 [103]) in recipient macrophages. These interactions activate the NFκB pathway and lead to the secretion of pro-inflammatory cytokines (IL-6 [102], TNF-α [102,103]) by recipient macrophages [102,103].”
149) EV-mediated progression of virus-associated disorders, Virus-EVs induced inflammatory disease, p10: See comment 83. Delete “Similarly, EBV encodes 49 mature miRNAs, that can all be secreted inside EVs [104]. During latent infection, EBV-EVs containing EBER1 miRNA and PS are recognized by theTIM-1 receptor of DCs. EBER1 uncapped 5’ triphosphate terminus is recognized by PRRs, leading to the activation of antiviral immune responses and consequential chronic inflammatory disease in individuals suffering from autoimmune disease [60].”
150) EV-mediated progression of virus-associated disorders, Virus-EVs induced inflammatory disease, p10: Delete “and mRNA”, and add reference in “HTLV-1 infected cells also secrete EVs containing viral Tax protein and mRNA and host proinflammatory molecules (GM-CSF, IL-6).”
151) EV-mediated progression of virus-associated disorders, Virus-EVs induced inflammatory disease, p10: Add references to support “Upon reception of Tax, recipient DCs secrete IL-10, IL-12, IL-17A, IFN-γ 506 and G-CSF, which could activate Th1, Th17 and cytotoxic T cells”
152) EV-mediated progression of virus-associated disorders, Virus-EVs induced inflammatory disease, p10: Justify. Add references to support “Virus-EV induced activation of PRRs in immune cells can also be achieved indirectly. HIV-EVs containing the Nef protein downregulate the ATP binding cassette transporter type A1 (ABCA1) in recipient macrophages, leading to a cascade of cellular modifications.”
153) EV-mediated progression of virus-associated disorders, Virus-EVs induced inflammatory disease, p10: Add references to support “Downregulation of ABCA1 reduces cholesterol efflux, resulting in inactivation of Cdc42, 514 decreased actin polymerization, and increased abundance of lipid rafts.”
154) EV-mediated progression of virus-associated disorders, Virus-EVs induced inflammatory disease, p10: Delete since it looks a bit speculative “This influences TLR-4 concentration in lipid rafts, potentiating ERK1/2 signaling and activating NLRP3 inflammasome and interleukin-1β (IL-1β) responses [106] (Figure 4A).”
155) EV-mediated progression of virus-associated disorders, Virus-EVs induced inflammatory disease, p11: Is it true for for all the EVs and for all the cells? Specify cells. Justify, and add references to support “Finally, uninfected cells can also secrete EVs that promote chronic inflammation.”
156) EV-mediated progression of virus-associated disorders, Virus-EVs induced inflammatory disease, p11: Add references to support “During DV infection, IL-1β is secreted as part of the antiviral immune response, which promotes secretion of EVs by non-infected platelets.”
157) EV-mediated progression of virus-associated disorders, Virus-EVs enhance liver disease, Title, p11: Replace EVs by extracellular vesicles.
158) EV-mediated progression of virus-associated disorders, Virus-EVs enhance liver disease, p11: Add references “HBV and HCV-EVs secreted by infected hepatocytes are integrated by hepatic stellate cells (HSCs), which cannot usually be infected by HBV and HCV, thus exasperating the disease to other cell types.”
159) EV-mediated progression of virus-associated disorders, Virus-EVs enhance liver disease, p11: Add references “HBV infected cells express the oncogenic viral protein HBx, which induces EV biogenesis by interacting with the cellular CD9, CD81 and neutral sphingomyelinase 2 (N-SMase).”
160) EV-mediated progression of virus-associated disorders, Virus-EVs enhance liver disease, p11: See comment 83. Delete “Likewise, HCV-EVs contain the upregulated host miR-19a RNA, which activates SOC3-STAT3-TGF-β pathway in HSCs, leading to fibrosis and worsened liver disease [109]”
161) EV-mediated progression of virus-associated disorders, Virus-EVs promote tumorigenesis, Title p11: Replace EVs by extracellular vesicles.
162) EV-mediated progression of virus-associated disorders, Virus-EVs promote tumorigenesis, p11: Add references to support Such viruses as the Human Papillomavirus (HPV), EBV, HIV-1, and HTLV-1 are widely known for their ability to cause cancer.”
163) EV-mediated progression of virus-associated disorders, Virus-EVs promote tumorigenesis, p11: Avoid generalization. Is it true for all the EVs? Justify and add references to support “Not surprisingly, EVs produced from cells infected with these viruses can also have oncogenic properties.”
164) EV-mediated progression of virus-associated disorders, Virus-EVs promote tumorigenesis, p11: Delete RNAs. Justify and add references to support “Virus-EVs can aid in the development of tumors by transferring viral oncogenes or RNAs.”
165) EV-mediated progression of virus-associated disorders, Virus-EVs promote tumorigenesis, p11: See comment 83. Delete. “For instance, HPV type 542 16 (HPV-16) infection increases the expression of more than 50 host microRNA (miRNA) 543 in infected cells, ten of which are upregulated and selectively packaged into EVs as a result of the E6/E7 oncogene. These miRNAs inhibit apoptosis and senescence while inducing proliferation in both infected and recipient cells [110,111]. In addition, keratinocytes transduced with E6 and E7 oncogenes secretes EVs containing E6 and E7 mRNA, resulting in the expression of these oncogenes in nearby cells [112]. HTLV-1 infected cells secrete EVs containing viral Tax, HBZ and Env mRNAs and the viral oncogenic Tax protein. These HTLV-1-EVs enhance survival of recipient PBMCs by protecting them from FAS-mediated apoptosis due to Tax-mediated up-regulation of pro-survival signaling molecules (AKT, Rb) and activation of the NFkB pathway. This supports HTLV-1 associated adult T-cell leukemia/lymphoma [105]”
166) EV-mediated progression of virus-associated disorders, Virus-EVs promote tumorigenesis, p11: Specify cancer. Justify and add references to support “Virus-EVs can also cause cancer cells to adopt a more aggressive and invasive phenotype. For example, EBV-infected cells express the viral latent membrane protein-1 555 (LMP-1), a constitutively active signaling protein mimicking CD40.”
167) EV-mediated progression of virus-associated disorders, Virus-EVs promote tumorigenesis, p11-12: See comment 83. Delete. “HIV-1-EVs secreted by T cells contains miR-155-5p which promotes expression of proinflammatory factors (IL-6, IL-8, TGF-β) and migration molecules (collagen type I, matrix metallopeptidase 2) in recipient cervical cancer cells. In this tumoral context, IL-6 induces cancer cell proliferation and inhibits apoptosis through STAT3. Moreover, miR-155-5p targets AT-rich interactive domain (ARID2) DNA binding protein, which inhibits migration of cervical cancer cells through the NFkB pathway. Altogether, these HIV-1-EVs promote cervical cancer proliferation 571 and invasion [117]”
168) EV-mediated progression of virus-associated disorders, Virus-EVs promote tumorigenesis, p12: Specify cancer. Is it true for all the virus-EVs? Justify and add references to support “Cancer escape from immune surveillance can also be facilitated by the transportation of viral regulatory factors in virus-EVs.”
169) EV-mediated progression of virus-associated disorders, Virus-EVs promote tumorigenesis, p12: This is the first time that plasma-derived vesicles was mentioned in the manuscript. Plasma-derived vesicles have not the same properties than small-sized vesicles. This illustrates the difficulty to extrapolate the findings in a general manner. Justify. Add references to support “HPV-16-infected keratinocytes expressing E7 oncogene secrete plasma-membrane-derived EVs which inhibit the adaptive immune response.”
170) V-mediated progression of virus-associated disorders, Virus-EVs promote tumorigenesis, p12: It is not the presence of PS but its localization on the external leaflet that is significant. All EVs have PS. Delete miR-BARTs. Rephrase “Lymphoma EBV-infected cells secrete EVs containing miR-BARTs and PS, which are integrated by monocytes. miR-BARTs enhance the upregulation of IL-10, TNF-α and Arginase 1 in active M1 monocytes, which support their phenotype switch into a regulatory M2-like phenotype and promote tumor growth and lymphoma severity [61].”
171) V-mediated progression of virus-associated disorders, Virus-EVs promote tumorigenesis, p12: It is bit speculative and insufficiently supported by explanations and experimental evidence. Delete “Similarly, gastric carcinoma EBV-infected cells secrete EVs targeting DCs and suppress their maturation, resulting in a worse prognosis for patients”
172) EVs and viruses as therapeutic tools, Title, p12: Replace EVs by extracellular vesicles in the title.
173) EVs and viruses as therapeutic tools, p12: Avoid extrapolation. Is it really the case for each virus and each EVs? Specify multiple pathologies. Why multiples? Rephrase, justify and add references to support “Ultimately, the relationships between EVs and viruses plays a integral role in the 586 spread and progression of multiple pathologies. ”
174) EVs and viruses as therapeutic tools, p12: The sentence is too broad. It is unclear if it is a positive or negative contribution. Specify pathologies. Justify, add references to support “A key contributor to these pathologies is the immune system”
175) EVs and viruses as therapeutic tools, p12: Focus on the facts and not on embellished stories. Does-it occurs for each type of virus-EVs ? Specify immunomodulatory mechanisms. Rephrase, justify and add references to support “Virus-EVs act as potent immunomodulatory molecules that can serve to either activate or suppress the immune system.”
176) EVs and viruses as therapeutic tools, p12: Specify inflammatory diseases, pathologies. Justify, and add references to support “This can consequently lead to an onset of chronic inflammatory diseases, or permit immune evasion and escape of other pathologies, such as cancer.”
177) EVs and viruses as therapeutic tools, p12: The sentence is too broad. Specify the alternative, justify and add references to support “However, the alternative can also be true”
178) EVs and viruses as therapeutic tools, p12: It is too broad. Avoid embellished stories. Does it occurs for all types of EVs? Specify interactions, strategy, and disorders. Delete “Virus-EV interactions can be a powerful immunotherapy strategy to target a variety of disorders.”
179) EVs and viruses as therapeutic tools, p12: Avoid gratuitous sentence. Delete “Both viruses and EVs have a wide range of therapeutic applications.”
180) EVs and viruses as therapeutic tools, p12: Avoid embellished stories. The cited references (120, 121, 122,123) are not clinical trials. The references 120, 121, and 122 are reviews, while the reference 123 is an experimental paper. In fact paper 123 concluded “However, as recently highlighted by the International Societies for EV (ISEV) and for Cellular Therapies (ISCT) (Börger et al., 2020), despite the urgency induced by the current pandemic, EV‐based therapeutic developments for COVID‐19 will have to meet as strong criteria of manufacturing processes, quality controls and compliance to safety regulation as any other therapies, before they can be implemented in human subjects.” Delete “EVs have reached clinical trials for a wide variety of different illnesses – from cancer [120] to cardiovascular disease, type 1 diabetes, neurodegenerative diseases (Huntington disease, Parkinson’s disease) [121], autoimmune and inflammatory diseases [122], and even for acute respiratory distress syndrome (ARDS) from SARS-CoV-2 infection [123].”
181) EVs and viruses as therapeutic tools, p12: Justify, and indicate clearly if there are in clinical trials “Similarly, viral vectors have been used for vaccine development for decades and are now being readily explored for gene therapies, vaccines and oncolytic viral therapies [124–127].”
182) EVs and viruses as therapeutic tools, p12: Justify and add references to support “However, EV and virus therapies both have inherent strengths and weaknesses.”
183) EVs and viruses as therapeutic tools, p12: I am not convinced that all EVs are safe and can cross all the barriers. Delete “EVs are safe, stable,have low immunogenicity and can intrinsically cross tissue and cellular barriers, but are challenging to produce and load efficiently with drugs [120,121].”
184) EVs and viruses as therapeutic tools, p12: Justify and add references to support “In this respect, combinational virus-EV therapeutics have many promising characteristics that overcome weaknesses of their individual counterparts.”
185) EVs and viruses as therapeutic tools, p12: Justify and add references to support “Hence, viral therapies that spread via EVs, either naturally or by design, have the potential of reaching more target cells without clearance from the immune system.”
186) EVs and viruses as therapeutic tools, p12: Justify and add references to support “Viruses can also help overcome low loading of therapeutic payloads in EVs by propagating in recipient cells and delivering prolonged expression of therapeutic transgenes.”
187) EVs and viruses as therapeutic tools, p12: Justify and add references to support “Virus infection is also known to promote EV production and secretion in many cells, so there is added potential for using engineered viruses with selectivity for different tissues to generate EVs with therapeutic payloads.”
188) EVs and viruses as therapeutic tools, Engineering viruses to target Evs, p12: Justify, and add references to support “Despite having an excellent safety profile and high efficiency in tissue transduction, there are still some drawbacks to using AAV for gene therapy.”
189) EVs and viruses as therapeutic tools, Engineering viruses to target Evs, p13: Add references to support “To overcome these issues, AAVs can be targeted to EVs to provide protection against neutralizing antibodies while keeping the ability to target specific tissues and limiting off-target infection.”
190) EVs and viruses as therapeutic tools, Engineering viruses to target Evs, p13: Add references to support “AAV-EVs have a high potential for the treatment of neurological disorders.”
191) EVs and viruses as therapeutic tools, Engineering viruses to target Evs, p13: Add references to support “AAV-EVs also showed promising results against hemophilia B genetic disease. Meiani et al. transfected HEK293 cells with AAV8 plasmids encoding the human coagulation factor IX, and later harvested AAV-EVs presenting TSG101 and CD9 specific markers.”
192) EVs and viruses as therapeutic tools, Engineering viruses to target Evs, p13: Add references to support “After intravenous injection, these AAV-EVs are protected from antibody recognition and successfully target and transduce liver cells, brain cells and skeletal muscle cells in mice. ”
193) EVs and viruses as therapeutic tools, Engineering viruses to target Evs, p13: Add references to support “Finally, AAV-EV strategies have also been studied for cancer therapy.”
194) EVs and viruses as therapeutic tools, Engineering viruses to target Evs, p13: Add references to support “Transfected HEK293 cells with AAV6 plasmids expressing luciferase have been tested for the effect of AAV-EVs on non-small cell lung cancer (carcinoma and adenocarcinoma) and small cell lung cancer cell lines.”
195) EVs and viruses as therapeutic tools, Natural transport of OVs in EVs enhances oncolytic viral therapy, Title, p13: Replace OVs and EVs by their full name in the title.
196) EVs and viruses as therapeutic tools, Natural transport of OVs in EVs enhances oncolytic viral therapy, p13: It is too broad. Specify relationships, EVs, viruses and immune response. Justify, and add references to support “The intertwined relationship between EVs, viruses, and activation of the immune response has a promising role in the development of oncolytic viral therapy.”
197) EVs and viruses as therapeutic tools, Natural transport of OVs in EVs enhances oncolytic viral therapy, p13: Add references to support “Oncolytic viruses (OVs) selectively infect tumor cells causing cell death, release of tumor and viral antigens and activation of an antiviral and anti-tumoral immune responses.”
198) EVs and viruses as therapeutic tools, Natural transport of OVs in EVs enhances oncolytic viral therapy, p13: References 138 and 140 are not clinical trials but reviews. Indicate references for the approbation by the .S Food and Drug Administration and the European Medicine Agency to support “For example, Talimogene laherparepvec (T-VEC) is a genetically modified HSV-1 encoding granulocyte-macrophage colony-stimulating factor (GM-CSF) that was approved by the U.S Food and Drug Administration and the European Medicine Agency in 2015 for the treatment of advanced melanoma [138,140].”
199) EVs and viruses as therapeutic tools, Natural transport of OVs in EVs enhances oncolytic viral therapy, p13-14: Specify that a Phase 2 trial (and not Phase trial 3, which could eventually lead to approval) was reported as stated in Ref 142. Rephrase “The triple mutant HSV-1 G47Δ OV (Delytact) has also received approval in Japan for treatment of malignant glioma and in clinical trials for treatment of prostate cancer, malignant pleural mesothelioma and reccurant olfactory neuroblastoma [141–143].”
200) EVs and viruses as therapeutic tools, Natural transport of OVs in EVs enhances oncolytic viral therapy, p14: The sentence is too broad. Delete “These EVs create an immunosuppressive environment for the tumor to grow or transfer oncogenes to healthy cells to promote tumor invasion. ”
201) EVs and viruses as therapeutic tools, Natural transport of OVs in EVs enhances oncolytic viral therapy, p14: Specify which EVs, which cancers, which cells to support “EVs can also promote cancer metastasis by enhancing cell migration, modulating the extracellular matrix and modifying stromal cells to prepare a 686 pre-metastatic niche [14].”
202) EVs and viruses as therapeutic tools, Natural transport of OVs in EVs enhances oncolytic viral therapy, p14: It is a too broad and a bit speculative. Delete “Because of these intense EVs exchanges within the tumor, and the capacity of OVs to induce EV secretion to either enhance viral spreading or increase the antiviral immune response, it is interesting to study whether EVs enhance virotherapy efficiency.”
203) EVs and viruses as therapeutic tools, Natural transport of OVs in EVs enhances oncolytic viral therapy, p14: Specify EVs and add references to support “Several studies have showed that EVs increase spreading of oncolytic adenoviruses, enhance the immune response, and help target metastatic niches.”
204) EVs and viruses as therapeutic tools, Natural transport of OVs in EVs enhances oncolytic viral therapy, p14: Specify EVs and add references to support “Many studies are thus turning toward EVs as vector for OV’s safe and specific transport to the tumor site.”
205) EVs and viruses as therapeutic tools, Engineering of OV-EV targeted therapies, Title, p14: Replace OVs and EVs by their full name in the title.
206) EVs and viruses as therapeutic tools, Engineering of OV-EV targeted therapies, p14: Delete “Despite their efficiency, many OVs face issues due to the already present anti-viral mmune response. OVs are also often delivered via intratumoral injection, which limits the efficiency of the treatment to metastatic tumors.”
207) EVs and viruses as therapeutic tools, Engineering of OV-EV targeted therapies, p14: Specify EVs, tissues, justify and add references to support “However, as mentioned above, EVs also have the ability to naturally uptake some OVs (OV-EVs) and improve delivery to target tissues.”
208) EVs and viruses as therapeutic tools, Engineering of OV-EV targeted therapies, p14: In the whole manuscript. There was insufficient information on the extraction, and characterizations of EVs. Add references and more information to support “EVs were produced by LLC1 murine Lewis lung carcinoma cell line.”
209) EVs and viruses as therapeutic tools, Engineering of OV-EV targeted therapies, p14: Specify tumor, add references to support “Ad-EV-Chemo induced T cells activation and infiltration inside the tumor.”
210) EVs and viruses as therapeutic tools, Engineering of OV-EV targeted therapies, p15: Delete since it is not directly related to viruses. “Similarly, plasma membrane-derived EVs can also carry chemotherapeutic drugs (cisplatin) [158] and oncolytic Ad (Ad5) [159] to the tumor site. Ad5-EVs secreted by A549 cancer cells are protected from antibody neutralization and deliver Ad5 inside tumor cells in a viral receptor independent manner. Moreover, this therapy also successfully target stem-like tumor repopulating cells, thus preventing cancer relapse [159]. Altogether, these studies support the potential of Ad-EVs and Ad-EVs- Chemo in cancer therapy, which can target tumor cells after intravenous injection due to their protection from neutralizing antibodies, eventually inducing T cell infiltration inside the tumor and decreasing tumor growth”
211) EVs and viruses as therapeutic tools, Engineering of OV-EV targeted therapies, p15: Justify and add references to support “OV-EVs can also be engineered using EV-mimetic nanovesicle drug loading technology, which allows production of a much higher quantity of EVs containing the drug of interest.”
212) EVs and viruses as therapeutic tools, Engineering of OV-EV targeted therapies, p15: It is unclear how cells produce drugs. Add explanations to support “Producer cells are first transduced to express a drug, and then squeezed step by step through smaller and smaller nanosized filters, eventually forcing the formation of EV-mimetic nanovesicles containing the drug [160]”
213) EVs and viruses as therapeutic tools, Packaging of therapeutic miRNAs into EVs via engineered OVs, Title, p15: Replace OVs and EVs by their full name in the title.
214) EVs and viruses as therapeutic tools, Packaging of therapeutic miRNAs into EVs via engineered OVs, p15: Justify, and add references. “Taken together, these findings support the development of virally-encoded, EV-delivered amiRNAs as a strategy to promote virus spread within tumours and modify the TME.”
215) Conclusion, p16: Delete “EV-mediated transfer of proteins, lipids or nucleic acids to recipient cells is essential in physiologic cell to cell communication, but also in viral infection and immune modulation. Both RNA and DNA viruses, whether they are enveloped or not, utilize the EVs pathway to secrete their viral particles, proteins, nucleic acids, but also to secrete host elements.”
216) Conclusions, p16: Specify cells, response, and add references to support “Virus-EVs enhance viral infection by transferring viral elements to recipient cells and modulating their response towards the virus.”
217) Conclusions, p16: It is a bit speculative. Delete “Moreover, virus-EVs modulate 795 many aspects of the immune system, leading to both antiviral and pro-viral responses that 796 can drive a variety of autoimmune and chronic inflammatory diseases and cancers.”
218) Conclusions, p16: Add specific examples, and references to support “EVs and viruses are both important vectors used in many therapies.”
219) Conclusion, p16: Add references and specific examples to support “In the field of oncology, 798 EVs-based therapies and virotherapies are both being developed.”
220) Conclusion, p16: Delete “Indeed, packaging of OV into EVs provides safe delivery and better uptake of the OV to the tumor, which enhances the overall efficacy of the virotherapy. ”
221) Conclusion, p16: Delete “Despite still being in development, this strategy has great potential in cancer therapy.”
Author Response
We would like to thank the reviewer for taking the time to review our manuscript and for offering a thorough critique with ways to improve the quality of the review.
I)The abstract was too broad and was unclear what were the key findings.
We wanted to have a general abstract that would grab the attention of readers and get them to read the entire review. Since this is a review there are also not much ‘key findings’ to highlight. Instead the review serves to discuss and summarize interesting connections between viruses and EVs. We therefore thing the current abstract fits with the scope of the review.
- II)Several sentences lacked justifications and references.
Regarding this and many of the minor comments below, we do not feel it is necessary to repeatedly cite references when we have introduced a specific topic or study (with corresponding references) and then proceed to discuss the details of those studies further.
III) The part concerning EVs carrying RNA shall be deleted. Recent experimental evidence indicated that the majority of extracellular RNA is associated with supermeres rather than small EVs and exomeres (1- Qin Zhang,Dennis K. Jeppesen, James N. Higginbotham,Ramona Graves-Deal,Vincent Q. Trinh,Marisol A. Ramirez,Yoojin Sohn,Abigail C. Neininger,Nilay Taneja,Eliot T. McKinley,Hiroaki Niitsu, Zheng Cao, Rachel Evans, Sarah E. Glass, Kevin C. Ray,William H. Fissell,Salisha Hill,Kristie Lindsey Rose,Won Jae Huh,Mary Kay Washington,Gregory Daniel Ayers,Dylan T. Burnette,Shivani Sharma,Leonard H. Rome,Jeffrey L. Franklin, Youngmin A. Lee,Qi Liu,and Robert J. Coffey. Supermeres are functional extracellular nanoparticles replete with disease biomarkers and therapeutic targets. Nat Cell Biol. 2021; 23(12): 1240–1254. 2- Tosar JP, Cayota A, Witwer K. Exomeres and supermeres: Monolithic or diverse? J. Extracellular Bio. 2021; 1:e45.).
Supermeres are still a class of extracellular vesicle. We have removed a line from our introduction (pg 2, line 52) that describes the EVs that we focused on throughout the review. That line was written during an early stage of the review before we expanded to a wider range of EVs.
- IV)There is a confusion between an enrichment of phosphatidylserine and its localization on the external leaflet of EVs, which appeared in several parts of the manuscript.
Thank you, this is described on page 9 line 433-435.
- V)Several findings were taken as a face value and were embellished. The overall manuscript lacked critical analysis.
We simply reported findings that have been published in peer-reviewed journals. As mentioned previously, our goal was to summarize and discuss the ways in which viruses use EVs and EV pathways during infection and virotherapy.
- VI)The manuscript appears as a list of facts, with no clear distinction between viruses, target cells. Not all the viruses behave in the same manner, and not all the cells have the same types of function. This yields to awkward generalizations which need to be corrected.
We have added 2 tables to help specifically indicate the effects that certain viruses have through EV-related mechanisms
VII) There was insufficient information on the properties of EVs. There was an insufficient discussion on the difficulties to extract a pure population of EVs. The possible variability of the properties of EVs was not discussed.
The production of EVs has been extensively discussed in a variety of reviews and papers. We chose to leave that out to focus more specifically on their relationships with viruses. We have indicated in the new “Table 2” how the EVs in those therapeutics were produced.
VIII) The conclusion was relatively broad, lacked justifications and references.
We wanted to use the conclusion provide a broad summary of the review. The statements in that section were discussed thoroughly throughout the review and since we use generalized statements, we do not feel that need references.
We wanted to thank the reviewer again for their helpful comments. While we have not responded here to the 200+ minor comments that they provided, we did carefully review them all and made changes to the text where we thought was necessary.
Reviewer 5 Report
In general, the author summarized the relationships between viruses and EVs and many developments in the engineering of virus-RV therapies. The review is very comprehensively
For this review, it would be better to summarize the information by tables for each part.
Author Response
In general, the author summarized the relationships between viruses and EVs and many developments in the engineering of virus-RV therapies. The review is very comprehensively
Thank you for the kind words and taking the time to review our manuscript.
For this review, it would be better to summarize the information by tables for each part.
We have added Tables for sections 4 and 6 (former sections 5 and 7) to help summarize some key information in the review
Round 2
Reviewer 4 Report
Answer point by point to all the minor concerns.